# Apoptotic signaling clears engineered *Salmonella* in an organ-specific manner

Taylor J Abele[1,2,3], Zachary P Billman[1,2,3,4], Lupeng Li[1,2,3], Carissa K Harvest[1,2,3,4], Alexia K Bryan[1,5], Gabrielle R Magalski[4], Joseph P Lopez[4], Heather N Larson[1,2,3], Xiao-Ming Yin[6], Edward A Miao[1,2,3]*

[1]Department of Integrative Immunobiology, Duke University School of Medicine, Durham, United States; [2]Department of Molecular Genetics and Microbiology, Duke University School of Medicine, Durham, United States; [3]Department of Cell Biology, Duke University School of Medicine, Durham, United States; [4]Department of Microbiology and Immunology, University of North Carolina at Chapel Hill, Chapel Hill, United States; [5]Department of Biomedical Engineering, Duke University Pratt School of Engineering, Durham, United States; [6]Department of Pathology and Laboratory Medicine, Tulane University School of Medicine, New Orleans, United States

*For correspondence:
edward.miao@duke.edu

**Abstract** Pyroptosis and apoptosis are two forms of regulated cell death that can defend against intracellular infection. When a cell fails to complete pyroptosis, backup pathways will initiate apoptosis. Here, we investigated the utility of apoptosis compared to pyroptosis in defense against an intracellular bacterial infection. We previously engineered *Salmonella enterica* serovar Typhimurium to persistently express flagellin, and thereby activate NLRC4 during systemic infection in mice. The resulting pyroptosis clears this flagellin-engineered strain. We now show that infection of caspase-1 or gasdermin D deficient macrophages by this flagellin-engineered *S.* Typhimurium induces apoptosis in vitro. Additionally, we engineered *S.* Typhimurium to translocate the pro-apoptotic BH3 domain of BID, which also triggers apoptosis in macrophages in vitro. During mouse infection, the apoptotic pathway successfully cleared these engineered *S.* Typhimurium from the intestinal niche but failed to clear the bacteria from the myeloid niche in the spleen or lymph nodes. In contrast, the pyroptotic pathway was beneficial in defense of both niches. To clear an infection, cells may have specific tasks that they must complete before they die; different modes of cell death could initiate these 'bucket lists' in either convergent or divergent ways.

## eLife assessment

Host cell death is an effective strategy to protect against infection, and is believed to function primarily by the elimination of the intracellular niche for pathogen replication. Abele and colleagues address an **important** question: does the mode of cell death affect its effectiveness in pathogen clearance? Consistent with prior observations, the authors provide **compelling** new evidence that the answer can depend on the cell type and/or tissue involved.

## Introduction

Regulated cell death (RCD) is essential to the survival of a species. There are many reasons why a cell may initiate death signaling, including DNA damage, disruptions to the cell cycle, or detection of an intracellular infection. Subsequently, there are several pathways through which a cell can accomplish RCD, including apoptosis and pyroptosis. These forms of RCD are essential in immune defense

**eLife digest** Although alive and healthy cells are essential for survival, in certain circumstances – such as when a cell becomes infected – it is beneficial for cells to deliberately die through a process known as regulated cell death. There are several types of regulated cell death, each with distinct pathways and mechanisms. However, if the initial pathway is blocked, cells can use an alternative one, suggesting that they can compensate for one other.

Two forms of regulated cell death – named pyroptosis and apoptosis – can be used by infected cells to limit the spread of pathogens. However, it was not clear if these two forms or additional 'back-up' apoptosis pathways – which are induced when pyroptosis fails – are equally efficient at clearing infections and how they might vary in different cell types.

To address this, Abele et al. investigated cell death in live mice infected with the bacterium *Salmonella*. Different organs in which the bacterium infects distinct cell types were examined. Experiments showed that pyroptosis could eliminate bacteria from both intestinal cells as well as immune cells found throughout the body, called macrophages. In contrast, apoptosis was only able to clear infection from intestinal cells.

The findings can be explained by prior studies showing both apoptosis and pyroptosis lead to the same outcome in intestinal cells – dead cells are expelled from the body through a process called extrusion to maintain the barrier function of the intestine. However, in macrophages, the different pathways lead to different outcomes, indicating they are not entirely interchangeable.

Overall, the findings of Abele et al. underscore the complexity of cellular responses to infection and the nuanced roles of different cell death pathways. This provides further evidence that cells might have specific tasks they need to complete before death in order to effectively clear an infection. These tasks may differ depending on cell type and the form of regulated cell death, and may not be equally efficient at clearing an infection.

against a variety of intracellular pathogens. Failing to accomplish cell death could allow a pathogen to replicate unchecked in a sequestered intracellular niche. On the surface, it might seem that terminating the intracellular replication niche is the most important task for RCD to accomplish. In this case, any modality of RCD should be sufficient to counteract an intracellular infection. That said, apoptosis and pyroptosis have distinct consequences.

Apoptosis is a non-lytic form of RCD that does not inherently cause inflammation. Apoptosis can be initiated via intrinsic or extrinsic signals (*Czabotar et al., 2014*; *Nagata and Tanaka, 2017*). The intrinsic pathway is regulated by the BCL-2 family of proteins, which monitor the cell for signs of damage or stress. Cellular stress leads to the activation of the pro-apoptotic subset of this family, called BH3-only proteins (e.g. BID). When these activation signals overcome inhibition by the anti-apoptotic subset of this family (e.g. BCL-2, MCL-1), it leads to assembly of BAK and/or BAX that form pores on mitochondria. The resulting mitochondrial outer membrane permeabilization (MOMP) leads to the release of cytochrome *c*. Cytochrome *c* initiates the formation of the apoptosome, activating caspase-9, which in turn cleaves and activates caspase-3, the prototypical executioner for apoptosis. Caspase-9 and –3 also activate caspase-7, which can cleave many of the caspase-3 target proteins that drive apoptosis, but whose unique function is to hyperactivate membrane repair (*Nozaki et al., 2022b*). Apoptosis can also be activated by cell-extrinsic pathways through death receptors (e.g. FAS, TNFR1), which in turn leads to the activation of caspase-8 (*Czabotar et al., 2014*; *Nagata and Tanaka, 2017*). These signaling pathways can reinforce each other, as caspase-8 can also cleave BID. Caspase-3 then cleaves numerous substrates that together convert a cell into apoptotic bodies.

In contrast, pyroptosis is a lytic, inherently inflammatory form of RCD typically initiated by inflammasomes, which detect signs of infection within the cytosolic compartment. Some inflammasomes directly detect microbial molecules, including proteins, LPS, or DNA; other inflammasomes indirectly detect the cytosolic perturbations caused by toxins (*Jorgensen et al., 2017*; *Nozaki et al., 2022a*). For example, NLRC4 can detect the activity of bacterial type III secretion systems when they aberrantly inject flagellin, rod, or needle proteins. These are directly detected via NAIP5/6, NAIP2, and NAIP1, respectively. NLRC4 can signal directly to caspase-1 to initiate pyroptosis, or indirectly via the ASC adaptor (encoded by *Pycard*). ASC facilitates enhanced caspase-1 activation and is necessary

for caspase-1 to efficiently cleave the cytokines pro-IL-1β and pro-IL-18 into their mature forms (*Broz et al., 2010b*). However, after NLRC4 activation, the direct activation of caspase-1 results in gasdermin D cleavage and activation without a requirement for ASC (*Broz et al., 2010b*; *Kovacs and Miao, 2017*). Cleaved gasdermin D then forms a pore on the cell membrane, which is large enough to release mature IL-1β and IL-18 from the cell and to cause pyroptosis (*Xia et al., 2021*).

While apoptosis and pyroptosis are caused by distinct signaling events, there is nevertheless cross-talk between these pathways. For example, there are distinct nodes where pyroptotic signaling proteins can also initiate apoptotic signaling. If downstream pyroptotic signaling is then inhibited, the cell will undergo apoptosis instead. Herein, we refer to these pathways as backup apoptosis. The most apical node is ASC, which can recruit caspase-8 (*Pierini et al., 2012*). Below this, caspase-1 can activate two different backup apoptosis pathways – by cleaving BID (*Guégan et al., 2002*; *Heilig et al., 2020*; *Tsuchiya et al., 2019*) and by cleaving caspase-7 (*Lamkanfi et al., 2008*; *Mahib et al., 2020*). Although we propose that the primary function of caspase-7 is to drive membrane repair (*Nozaki et al., 2022b*), when it is activated in the absence of pores it might result in a partial form of apoptosis. All of this raises the question: are apoptosis and pyroptosis equal and redundant by simply eliminating the intracellular niche?

Many intracellular bacteria have evolved mechanisms to evade detection by inflammasomes (*Jorgensen et al., 2017*; *Lacey and Miao, 2020*; *Maltez and Miao, 2016*). Therefore, backup apoptosis could restore cell death in the case of bacteria that inhibit pyroptosis (*Luchetti et al., 2021*). To evade detection by inflammasomes, *Salmonella enterica* serovar Typhimurium encodes two distinct T3SS apparati. The *Salmonella* pathogenicity island 1 (SPI1) T3SS is used to invade epithelial cells. However, the flagellin that is co-expressed is detected by NLRC4, as are the SPI1 rod and needle proteins, albeit less efficiently (*Jorgensen et al., 2017*; *Pereira et al., 2011*; *Zhang et al., 2015*). This detection is protective for the host in a mouse gastroenteritis model by reducing *S.* Typhimurium intracellular burdens (*Broz et al., 2010a*). Once *S.* Typhimurium has successfully invaded a cell and occupies the *Salmonella*-containing vacuole (SCV), it represses SPI1 and instead expresses SPI2. The SPI2 T3SS evades detection by NLRC4 via mutations on its rod proteins and by repressing flagellin expression. This evasion permits intracellular replication in epithelial cells and macrophages (*Broz et al., 2010a*).

We previously demonstrated that during this SPI2-driven intracellular phase, *S.* Typhimurium can be engineered to activate NLRC4. This was accomplished by expressing the NLRC4 agonist FliC under a SPI2 promoter, herein called FliC$^{ON}$ (*Miao et al., 2011*; *Miao et al., 2010a*). FliC$^{ON}$ *S.* Typhimurium are detected by NLRC4 during the intracellular phase both in vitro and in vivo. This detection results in the clearance of FliC$^{ON}$ during systemic mouse infection (*Jorgensen et al., 2017*; *Jorgensen and Miao, 2015*; *Miao et al., 2010a*). This illustrates that the reason why *S.* Typhimurium normally represses flagellin is that pyroptosis is extremely detrimental to the bacterium during intracellular replication. Furthermore, FliC$^{ON}$ *S.* Typhimurium is useful as an experimental tool with which to investigate the effectiveness of pyroptosis in vivo. At the time of our prior publications, the backup apoptotic pathways were not yet discovered (*Heilig et al., 2020*; *Motani et al., 2010*; *Tsuchiya et al., 2019*). Here, we investigate whether apoptotic pathways could be useful in clearing intracellular infection. Because *S.* Typhimurium likely evades apoptotic pathways, we again use engineering in order to create strains that will induce apoptosis. This allows us to study apoptosis in a controlled manner in vivo.

## Results

### FliC$^{ON}$ *S.* Typhimurium activates apoptotic backup pathways in vitro

We first investigated the relative contribution of these backup pathways (*Figure 1A*) during FliC$^{ON}$ infection by examining in vitro infections in bone marrow-derived macrophages (BMMs) from various knockout mice. We wished to focus on the intracellular SPI2-driven phase of infection. To ensure SPI1-genes were not expressed, we grew *S.* Typhimurium in SPI2 inducing media prior to infection (*Miao et al., 2002*). As an additional measure, we used *S.* Typhimurium that were deficient for flagellin and SPI1 (*flgC* and *ΔprgH-K*) so that NLRC4 could only be activated by the engineered FliC$^{ON}$ system (*Figure 1B*). As a negative control, *Nlrc4*$^{-/-}$ BMMs did not activate either pyroptotic or backup apoptotic pathways (*Figure 1C–D*). *Casp1*$^{-/-}$ BMMs retain the ASC to caspase-8 pathway; and indeed, we observed weak caspase-8, BID, and caspase-3 cleavage at 4 hr post-infection (hpi) (*Figure 1D*),

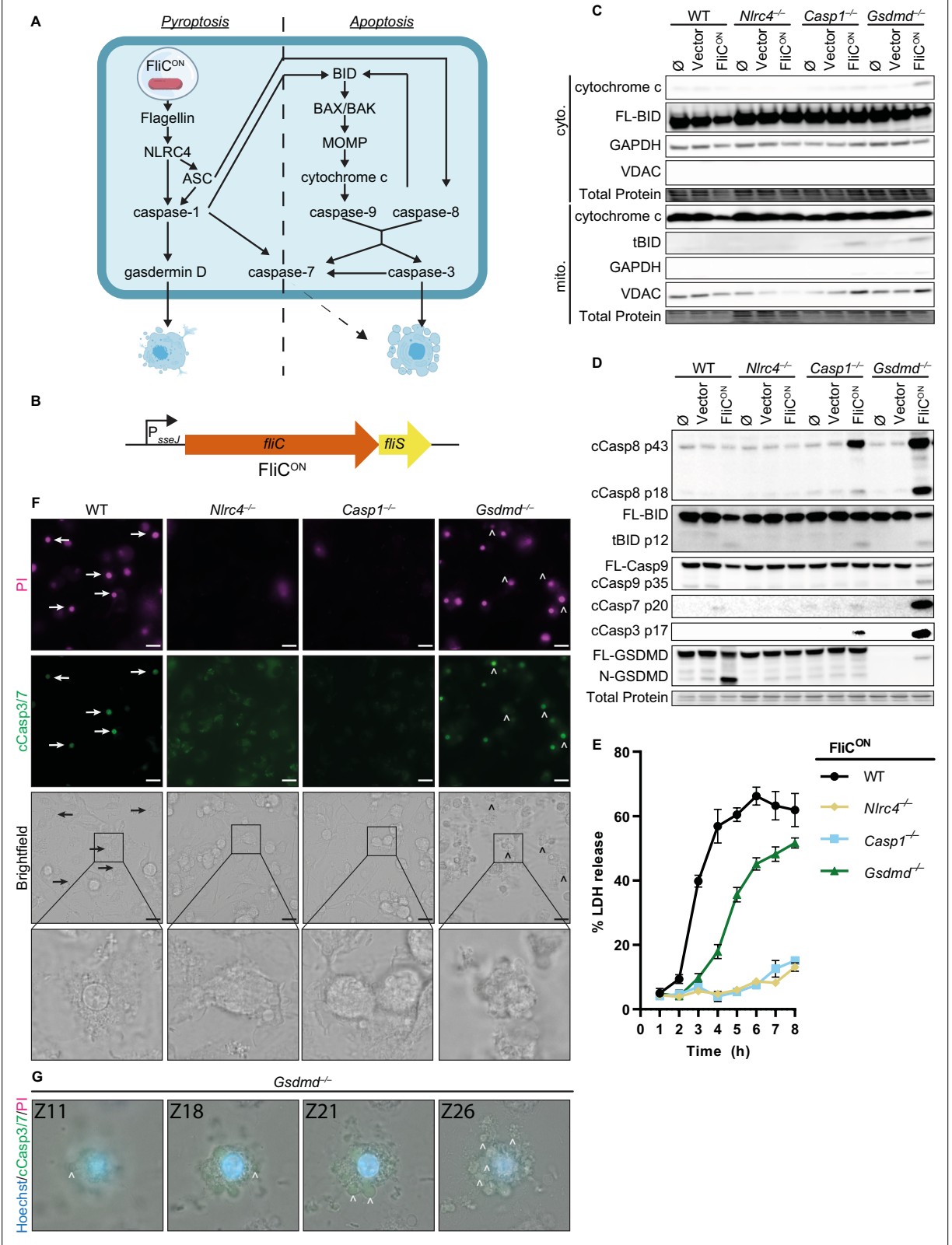

**Figure 1.** FliC[ON] *S.* Typhimurium activates apoptotic backup pathways in vitro. (**A**) Cell death pathways activated by FliC[ON] *S.* Typhimurium. (**B**) Schematic of engineered FliC[ON] construct. (**C–F**) Bone marrow-derived macrophages (BMMs) were infected with indicated SPI2-induced *S.* Typhimurium strains. (**C**) Western blot analysis of cytosolic and mitochondrial fractions at 5 hpi. Representative image from three independent experiments. (**D**) Western blot analysis of whole cell lysates at 4 hpi. Representative image from three independent experiments. (**E**) Lactate dehydrogenase (LDH) release at 1–8 hpi.

*Figure 1 continued on next page*

eLife Research article | Immunology and Inflammation | Microbiology and Infectious Disease

*Figure 1 continued*

Results representative of three independent experiments. Data are represented as mean ± SD of three technical replicates. (**F–G**) Immunofluorescence and brightfield. Cells were stained with PI, cleaved caspase-3/7, and Hoechst. (**F**) Representative image from two (brightfield, PI) or one (cleaved caspase-3/7) independent experiments at 4 hpi. 60 x magnification, scale bar 20 µm. Arrows, pyroptotic cells. Carrots, apoptotic cells. (**G**) Z-stack slices from *Figure 1—video 1*. *Gsdmd⁻/⁻* BMMs infected with FliC^ON imaged at 6 hpi. Representative Z-stack from three (brightfield, PI) or one (cleaved caspase-3/7) independent experiments. 60 x magnification, Z-slices 11, 18, 21, and 26 shown. Carrots; selected examples of apoptotic bodies.

The online version of this article includes the following video, source data, and figure supplement(s) for figure 1:

**Source data 1.** Western blot images for *Figure 1C*.

**Source data 2.** Western blot images for *Figure 1D*.

**Source data 3.** Data for *Figure 1E*.

**Figure supplement 1.** Vector control *S.* Typhimurium does not activate apoptotic backup pathways in vitro.

**Figure supplement 1—source data 1.** Western blot images for *Figure 1—figure supplement 1A*.

**Figure supplement 1—source data 2.** Data for *Figure 1—figure supplement 1B–C*.

**Figure 1—video 1.** Engineered FliC^ON *S.* Typhimurium activates apoptosis in *Gsdmd⁻/⁻* bone marrow-derived macrophages (BMMs) in vitro.
https://elifesciences.org/articles/89210/figures#fig1video1

indicating that this pathway is relatively slow. In contrast, *Gsdmd⁻/⁻* BMMs remain competent for both the ASC to caspase-8 and the caspase-1 to BID pathways; and indeed, we observed stronger cleavage of BID, caspase-9, and caspase-3 (*Figure 1D*). We also noticed stronger caspase-8 cleavage in the *Gsdmd⁻/⁻* BMMs compared to the *Casp1⁻/⁻* BMMs (*Figure 1D*), perhaps due to the ability of active caspase-3 and –7 to cleave caspase-8, in a feed-forward loop (*Inoue et al., 2009*; *McComb et al., 2019*). Further, only *Gsdmd⁻/⁻* BMMs displayed release of cytochrome *c* from the mitochondria to the cytosol during infection (*Figure 1C*), as expected by faster signaling from caspase-1 to BID. Indeed, we also observed cleaved BID (tBID) localization to the mitochondria (*Figure 1C*). Compared to *Gsdmd⁻/⁻* BMMs, *Pycard/Gsdmd⁻/⁻* BMMs and *Bid/Gsdmd⁻/⁻* BMMs had slightly reduced cleavage of caspase-8,–9, and –3 (*Figure 1—figure supplement 1A*) as expected due to the loss of only one of the two apoptotic pathways in each double knockout. Interestingly, WT BMMs, which are competent for all apoptotic and pyroptotic pathways showed strong cleavage of gasdermin D but only showed weak cleavage of BID and caspase-7, and lacked cleavage of caspase-9 and –3 (*Figure 1D*, *Figure 1—figure supplement 1A*), suggesting that completion of pyroptosis precludes the apoptotic signaling pathway.

We next assessed viability of WT, *Nlrc4⁻/⁻*, *Casp1⁻/⁻*, and *Gsdmd⁻/⁻* BMMs during FliC^ON infection using a lactate dehydrogenase (LDH) assay (*Figure 1E*, *Figure 1—figure supplement 1B*). WT BMMs showed rapid release of LDH consistent with pyroptosis. In accordance with our western blot data (*Figure 1C–D*), *Nlrc4⁻/⁻* BMMs infected with FliC^ON did not show toxicity over time (*Figure 1E*). While *Casp1⁻/⁻* BMMs did show weak caspase-3 cleavage by 4–6 hpi (*Figure 1D*), they do not show much LDH release over the course of 8 hpi (*Figure 1E*). As before, *Gsdmd⁻/⁻* BMMs showed early signs of loss of viability when compared to *Casp1⁻/⁻* BMMs (*Figure 1E*), agreeing with prior publications with different agonists (*Doerflinger et al., 2020*; *Tsuchiya et al., 2019*). *Pycard/Gsdmd⁻/⁻* BMMs and *Bid/Gsdmd⁻/⁻* BMMs showed reduced LDH release compared to *Gsdmd⁻/⁻* BMMs (*Figure 1E*, *Figure 1—figure supplement 1C*). Vector control infections showed no LDH release over time (*Figure 1—figure supplement 1B–C*). These data indicate that the several backup pathways that remain in a *Gsdmd⁻/⁻* BMM cause LDH release, albeit slower than pyroptosis.

Finally, we used propidium iodide (PI) and cleaved caspase-3/7 staining to visualize pyroptotic and apoptotic cells in vitro (*Figure 1F* and *Figure 1—figure supplement 1D–E*). As expected, both WT and *Gsdmd⁻/⁻* BMMs infected with FliC^ON for 4 hr showed strong PI staining, whereas neither *Nlrc4⁻/⁻* nor *Casp1⁻/⁻* BMMs had a high number of PI positive cells (*Figure 1F*, *Figure 1—figure supplement 1E*). Brightfield images of BMMs infected with FliC^ON showed clear pyroptotic morphology in WT BMMs, whereas *Gsdmd⁻/⁻* BMMs showed clear apoptotic blebbing that co-stained for cleaved caspase-3 (*Figure 1F*, *Figure 1—figure supplement 1E*). These apoptotic bodies are more clearly visualized in the form of a Z-stack (*Figure 1—video 1*, *Figure 1G*). *Casp1⁻/⁻* BMMs cellular morphology showed early signs of rounding but no apoptotic bodies, whereas *Nlrc4⁻/⁻* BMMs infected with FliC^ON have no difference in cellular morphology from the controls (*Figure 1F*, *Figure 1—figure supplement*

*1D*). Therefore, engineered FliC[ON] *S.* Typhimurium activates the apoptotic backup pathways in vitro in BMMs.

## Backup apoptosis fails to clear FliC[ON] *S.* Typhimurium in the spleen

We next investigated whether the apoptotic backup pathways cleared *S.* Typhimurium during in vivo systemic infection in mice. We used a competitive index infection model (*Beuzón and Holden, 2001*), where a single mouse is infected with a vector control WT *S.* Typhimurium mixed with FliC[ON] *S.* Typhimurium at a 1:1 ratio, each marked with a different antibiotic resistance (*Figure 2A* and *Figure 2—figure supplement 1A–B*). We use this competitive index method throughout this study. FliC[ON] was progressively cleared in WT mice at a rate of ~10 fold per day compared to WT *S.* Typhimurium (*Figure 2B–C*), consistent with our prior publications that pyroptosis is sufficient to clear FliC[ON] (*Jorgensen et al., 2016b*; *Miao et al., 2010a*). FliC[ON] was not cleared in *Nlrc4*[-/-] mice, as expected (*Figure 2B and D*). However, FliC[ON] was also not cleared in *Casp1*[-/-] mice (*Figure 2B and E*), even though they are sufficient for the ASC to caspase-8 apoptotic backup pathway (*Miao et al., 2010a*). Furthermore, FliC[ON] was not cleared in *Gsdmd*[-/-] mice that retain caspase-1 to BID, caspase-1 to caspase-7, and ASC to caspase-8 backup pathways (*Figure 2B and F*). Finally, FliC[ON] was not cleared in *Pycard/Gsdmd*[-/-] or *Bid/Gsdmd*[-/-] mice that retain fewer backup pathways (*Figure 2—figure supplement 1C–D*). This lack of clearance also held true in a higher dose 10[5] total CFU infection (*Figure 2G–H*). When examining only the burdens of the vector control, we observed trending, but not statistically significant increases in *Casp1*[-/-] and *Gsdmd*[-/-] mice, which may reflect incomplete evasion of NLRC4 and NLRP3 (*Broz et al., 2010a*; *Man et al., 2014*), but this detection of vector-containing WT *S.* Typhimurium is much less effective in clearance compared to FliC[ON] engineered bacteria (*Figure 2C–F*). Therefore, the apoptotic backup pathways are not sufficient to clear either a high or low dose *S.* Typhimurium infection from the spleen when pyroptotic signaling is blocked.

## Engineered BID[ON] *S.* Typhimurium activates apoptosis in vitro

Apoptosis is more commonly activated by intrinsic cellular signaling than backup pathways branching from pyroptosis. To trigger intrinsic apoptosis, we engineered *S.* Typhimurium to directly induces BID-dependent apoptosis. We fused the pro-apoptotic BH3 domain of murine BID to the T3SS secretion signal of SspH1 via an HA tag (*Figure 3A*). This SspH1[SS]-HA-BID[BH3] protein was expressed from a SPI2 promoter; we refer to these bacteria as BID[ON] (*Figure 3A*). The BH3 domain of BID is sufficient to activate BAX/BAK to cause MOMP, which should lead to intrinsic apoptosis (*Figure 3B*). BID[ON] bacteria grew normally in LB media (*Figure 3—figure supplement 1A*). In WT BMMs, BID[ON] caused caspase-3 cleavage, while a negative vector control without the BID[BH3] domain did not (*Figure 3C*). The weaker HA band in SspH1[SS]-HA-BID[BH3] compared to SspH1[SS]-HA could be due to caspase-3 cleaving after aspartates within the HA tag, which can efficiently destroy the HA epitope. The fused SspH1[SS]-HA-BID[BH3] protein is detectable by western blot with both α-HA and α-BID antibodies (*Figure 3C–D*). Coincidentally, SspH1[SS]-HA-BID[BH3] is similar in length to full-length endogenous BID (predicted 21 kDa and 20 kDa, respectively), and resolved as a thicker band or sometimes as a doublet at ~20 kDa (*Figure 3D–E*, *Figure 3—figure supplement 1B*). BID[ON] induced release of cytochrome *c* from the mitochondria, followed by weak caspase-9 cleavage, resulting in both cleaved caspase-3 and –7, and weak caspase-8 cleavage (*Figure 3E–F*). As expected, BID[ON] did not result in cleaved gasdermin D (*Figure 3E*), indicating that these timepoints are too early to observe the inactivating cleavage of gasdermin D by caspase-3 (*Chen et al., 2019*). Morphologically, BID[ON] infected BMMs showed classic apoptotic blebbing between 4–6 hpi (*Figure 3G*), which is best visualized via a Z-stack (*Figure 3—video 1*, *Figure 3H*). Therefore, *S.* Typhimurium can be engineered to trigger cell intrinsic apoptosis in vitro in BMMs.

## Apoptosis is induced slower than pyroptosis

Above we described two engineered strains, FliC[ON] and BID[ON], which can be used to initiate three different cell death pathways: (1) FliC[ON] in WT BMMs causes pyroptosis, (2) FliC[ON] in *Gsdmd*[-/-] BMMs causes backup apoptosis, and (3) BID[ON] in WT BMMs causes intrinsic apoptosis. We compared the kinetics and signaling of these three models to determine whether the different engineering methods induced cell death at different rates. We detected gasdermin D cleavage from FliC[ON] infection in as little as 1–2 hpi (*Figure 4A*). However, cleaved caspase-3 from FliC[ON] in *Gsdmd*[-/-] BMMs was not

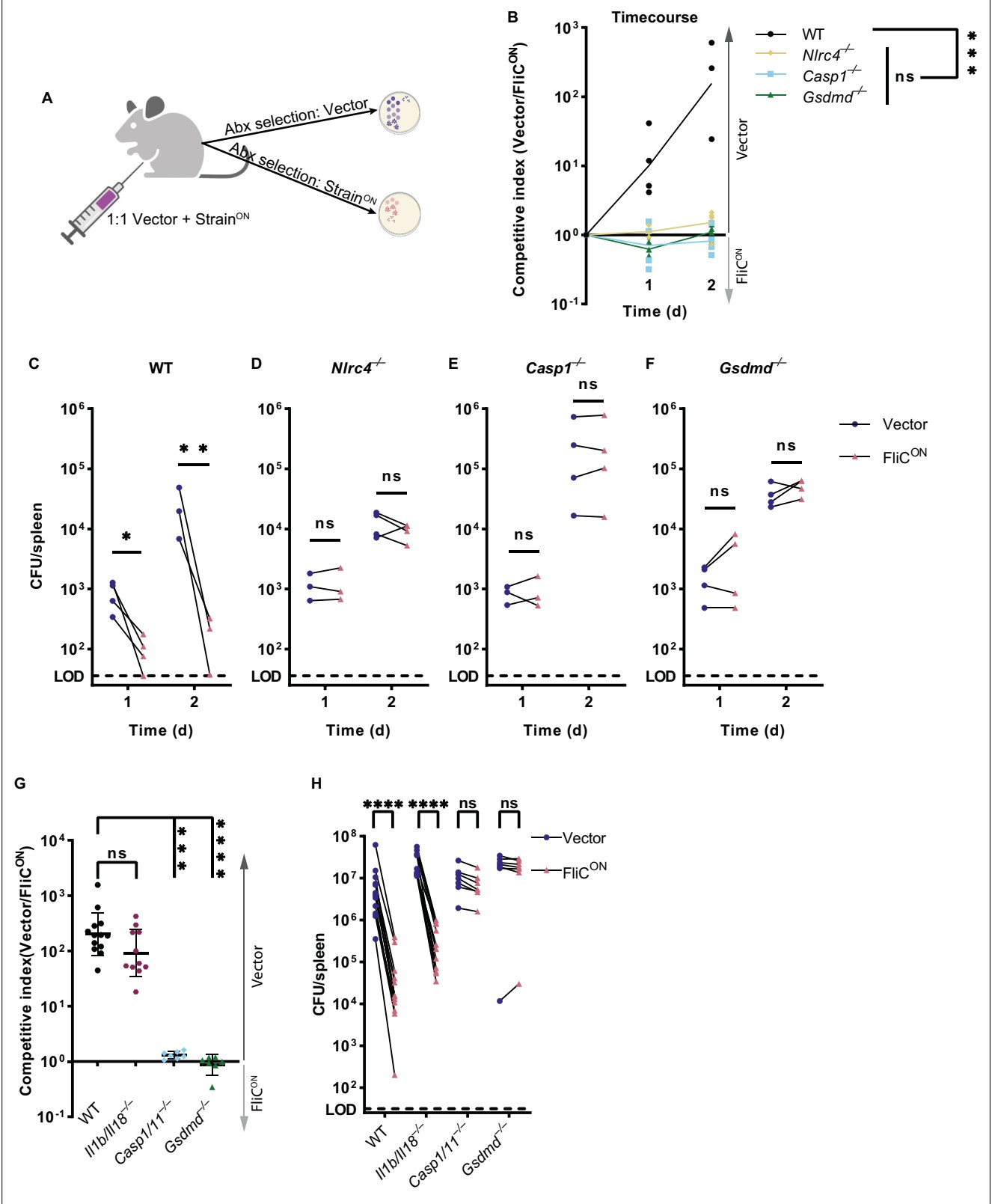

**Figure 2.** Backup apoptosis does not clear FliC[ON] *S.* Typhimurium in the spleen. (**A**) Schematic of competitive index infection model. (**B–H**) Mice were infected with a 1:1 ratio of FliC[ON] and a vector control *S.* Typhimurium. Bacterial burdens in the spleen were determined at the indicated timepoints. (**B**) Timecourse of competitive index infection in indicated mice. Mice were infected with $5 \times 10^2$ CFU of each strain. Ratio of vector to FliC[ON] is graphed. Data representative of three (WT, *Gsdmd*[−/−]) or two (*Nlrc4*[−/−], *Casp1*[−/−]) independent experiments. Line connects mean, n=3–4 mice per genotype per

*Figure 2 continued on next page*

*Figure 2 continued*

timepoint. Two-way ANOVA n.s. p>0.05; ***p<0.001. (**C–F**) Individual burdens of vector and FliC$^{ON}$ from (**B**). Paired vector and FliC$^{ON}$ data from each mouse are connected by a line. Two-way repeated measure ANOVA. n.s. p>0.05, *p<0.05, **p<0.01 (**G**) Mice were infected with $5 \times 10^4$ CFU of each strain. Bacterial burdens in the spleen were determined at 48 hpi. Ratio of vector to FliC$^{ON}$ is graphed. Combined two independent experiments, line representing mean ± SD, n=7–13 mice per genotype. Kruskal-Wallis n.s. p>0.05; ***p<0.001, ****p<0.0001. (**H**) Individual burdens from (**G**). Paired vector and FliC$^{ON}$ data from each mouse are connected by a line. Two-way repeated measure ANOVA. n.s. p>0.05, ****p<0.0001.

The online version of this article includes the following source data and figure supplement(s) for figure 2:

**Source data 1.** Data for *Figure 2B–H*.

**Figure supplement 1.** Competitive index model can be used to study the clearance of *S.* Typhimurium in vivo.

**Figure supplement 1—source data 1.** Data for *Figure 2—figure supplement 1A–D*.

detected until 3 hpi, and was pronounced only after 4 hpi (*Figure 4A*). BID$^{ON}$ induced apoptosis slower still, with cleaved caspase-3 only weakly detected at 4 hpi, and not pronounced until 5 or 6 hpi (*Figure 4A*). The difference between the kinetics of these apoptotic pathways is likely due to the efficient detection of FliC by the NAIP/NLRC4 inflammasome, which has a theoretical sensitivity to detect a single flagellin protein molecule (*Hu et al., 2015*; *Zhang et al., 2015*). On the other hand, BID$^{BH3}$ signaling is buffered by the cellular BCL family proteins, which are expected to delay the initiation of the BAK/BAX pore. Overall, both ways of achieving apoptosis are successful in vitro, but with slightly different kinetics.

These kinetics were also apparent visually, as we observed pyroptotic cells in WT BMMs infected with FliC$^{ON}$ at 2 hpi, whereas we did not see any apoptotic blebbing or cleaved caspase-3/7 in *Gsdmd$^{-/-}$* BMMs infected with FliC$^{ON}$ or BMMs infected with BID$^{ON}$ until 6 hpi (*Figure 4B*, *Figure 4—figure supplement 1A–B*, *Figure 1—video 1*, *Figure 3—video 1*). Correspondingly, we observed strong PI staining in WT BMMs infected with FliC$^{ON}$ at both 2 and 6 hpi (*Figure 4B*, *Figure 4—figure supplement 1B*). In contrast, *Gsdmd$^{-/-}$* BMMs infected with FliC$^{ON}$ showed no PI staining at 2 hpi, and were only PI positive later at 6 hpi. BID$^{ON}$ infected BMMs were similarly delayed, with slightly weaker PI staining (*Figure 4B*), again consistent with somewhat slower kinetics in the BID$^{ON}$ infection.

## Intrinsic apoptosis does not clear engineered *S.* Typhimurium in the spleen

Next, we wanted to determine whether BID$^{ON}$ was cleared in vivo using the competitive index assay, where, again, mice are co-infected with engineered bacteria and a vector control. Whereas FliC$^{ON}$ was cleared in WT mice, BID$^{ON}$ was not (*Figure 5A–B*). FliC$^{ON}$ clearance is progressive over 48 hpi, whereas BID$^{ON}$ burdens increased in ratios equivalent to vector control *S.* Typhimurium over this period (*Figure 5B*). We also compared our two apoptosis-inducing models, and found that both FliC$^{ON}$ in *Gsdmd$^{-/-}$* mice and BID$^{ON}$ in WT mice were statistically equal in their lack of clearance (*Figure 5C–D*).

Pyroptosis is pro-inflammatory, whereas apoptosis itself does not inherently promote inflammation. We hypothesized that specific inflammatory signals such as IL-1β, IL-18, or other cytosolic molecules released by pyroptosis might be required to promote bacterial clearance after RCD. These would be produced during FliC$^{ON}$ infection in WT mice, but should be absent during BID$^{ON}$ infection. To include pyroptosis-induced inflammatory signals during a BID$^{ON}$ infection, we created a triple competitive index model, wherein a single mouse is infected with equal ratios of vector control, FliC$^{ON}$, and BID$^{ON}$ simultaneously (*Figure 5E*, *Figure 5—figure supplement 1A–B*). We show that even in the presence of FliC$^{ON}$, BID$^{ON}$ cannot be cleared from the spleen to the same degree as FliC$^{ON}$ (*Figure 5F–G*). Therefore, FliC$^{ON}$ does not create a proinflammatory environment in the spleen that alters the failure of BID$^{ON}$ to be cleared. We noted a slight reduction in BID$^{ON}$ burdens compared to the vector control in this triple competitive index model that is likely an early manifestation of non-specific attenuation due to *S.* Typhimurium engineering that will be described in another manuscript (*Abele et al., 2023*). This suggests that the ability of RCD to lead to clearance of engineered *S.* Typhimurium is linked to the specific mode of RCD for that host cell.

## Pyroptosis clears FliC$^{ON}$ from myeloid cells

In the spleen, *S.* Typhimurium is established to primarily infect macrophages (*Salcedo et al., 2001*). To validate that clearance after induction of pyroptosis, but not backup apoptosis, was coming from the

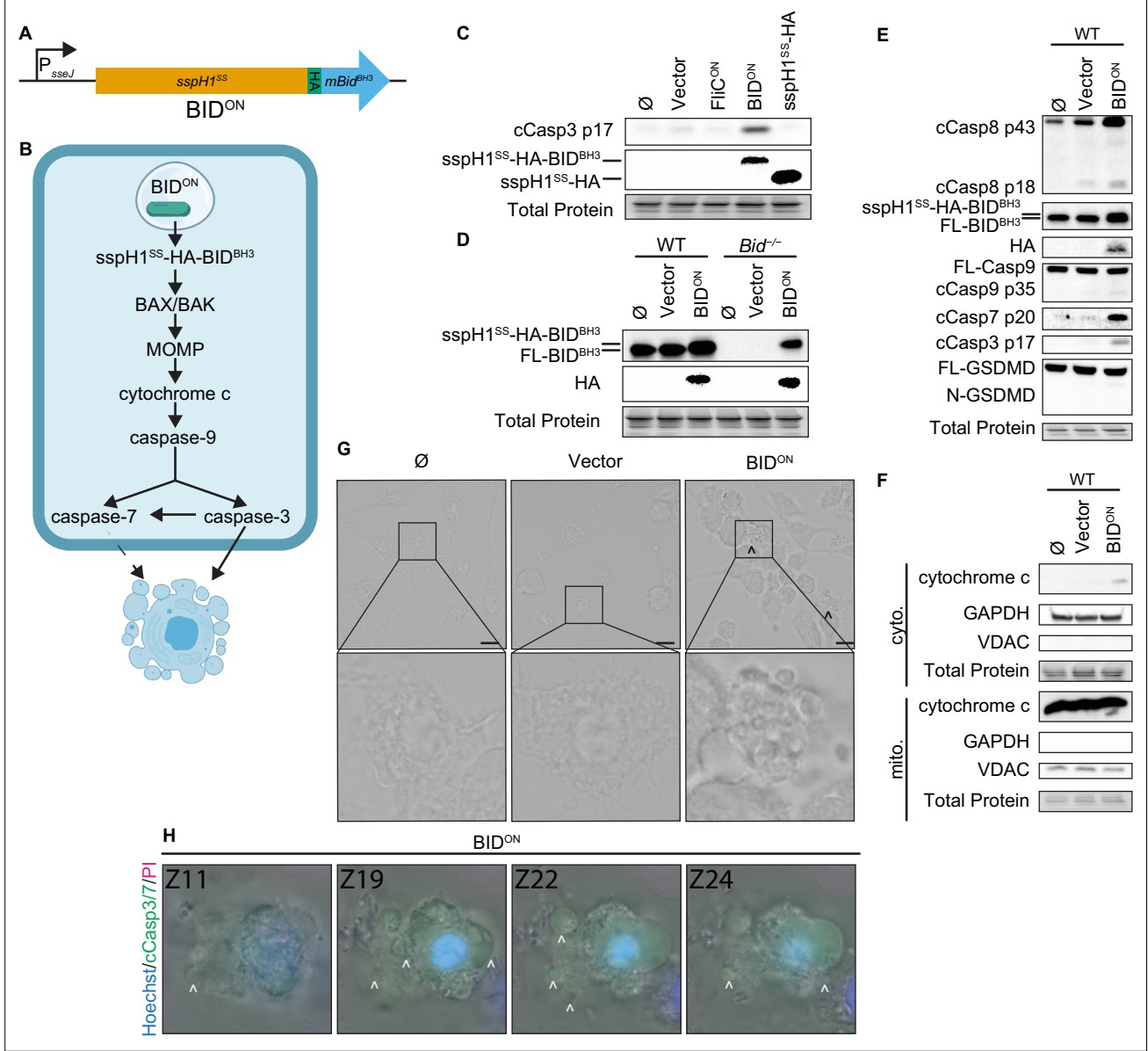

**Figure 3.** Engineered BID^ON *S.* Typhimurium activates apoptosis in vitro. (**A**) Schematic of engineered BID^ON construct. (**B**) Pathway model showing how BID^ON leads to intrinsic apoptosis. (**C–G**) Bone marrow-derived macrophages (BMMs) were infected with indicated SPI2-induced *S.* Typhimurium strains. (**C–E**) Western blot analysis of whole cell lysates at 6 hpi. Data representative of two (**C**) or three (**D–E**) independent experiments. (**F**) Western blot analysis of cytosolic and mitochondrial fractions at 4 hpi. Data representative from three independent experiments. (**G**) Brightfield at 6 hpi. Data representative of three independent experiments. 60 x magnification, scale bar 20 µm, carrot, apoptotic blebs. (**H**) Z-stack slices from *Figure 3—video 1*. WT BMMs infected with BID^ON imaged at 6 hpi. Representative Z-stack from three (brightfield, PI) or one (cleaved caspase-3/7) independent experiments. 60 x magnification, Z-slices 11, 19, 22, and 24 shown. Carrots; selected examples of apoptotic bodies.

The online version of this article includes the following video, source data, and figure supplement(s) for figure 3:

**Source data 1.** Western blot images for *Figure 3C*.

**Source data 2.** Western blot images for *Figure 3D*.

**Source data 3.** Western blot images for *Figure 3E*.

**Source data 4.** Western blot images for *Figure 3F*.

**Figure supplement 1.** Production of SspH1^SS-HA-BID^BH3 construct does not cause growth defects in BID^ON *S.* Typhimurium.

**Figure supplement 1—source data 1.** Data for *Figure 3—figure supplement 1A*.

**Figure supplement 1—source data 2.** Western blot images for *Figure 3—figure supplement 1B*.

*Figure 3 continued on next page*

*Figure 3 continued*

**Figure 3—video 1.** Engineered BID<sup>ON</sup> *S.* Typhimurium causes apoptosis in WT bone marrow-derived macrophages (BMMs) in vitro. https://elifesciences.org/articles/89210/figures#fig3video1

macrophage compartment, we infected cell-type specific *Casp1*<sup>fl/fl</sup> mice. We crossed these mice with either *Lyz2-cre* (also known as *LysM-cre*), which deletes caspase-1 from myeloid cells, or *Mrp8-cre*, which deletes caspase-1 efficiently from neutrophils and does not efficiently delete caspase-1 in monocytes and macrophages (*Abram et al., 2014*). We compared these cell-type-specific knockouts to caspase-1 sufficient littermate controls. We again used a competitive index model, and found that *Casp1*<sup>fl/fl</sup> *Mrp8-cre* mice retained the ability to clear FliC<sup>ON</sup>, whereas *Casp1*<sup>fl/fl</sup> *Lyz2-cre* mice lost the ability to clear FliC<sup>ON</sup> (*Figure 6A–B*). This is in contrast to our prior data, where we show that caspase-11 within the neutrophil compartment is necessary for clearance of *Burkholderia thailandensis* (*Kovacs et al., 2020*). Therefore, when macrophages have caspase-1, FliC<sup>ON</sup> *S.* Typhimurium is successfully cleared from the spleen. However, when caspase-1 is deleted from macrophages, FliC<sup>ON</sup> is no longer cleared. These data indicate that pyroptosis clears FliC<sup>ON</sup> from the macrophage compartment, whereas backup apoptosis fails to clear FliC<sup>ON</sup> from this compartment.

## Apoptotic pathways clear engineered *S.* Typhimurium during intestinal infection

The apoptotic backup pathways have previously been shown to successfully clear non-engineered WT *S.* Typhimurium infection downstream of NLRC4 in intestinal epithelial cells (IECs) (*Rauch et al., 2017*). This is in disagreement with our results above using engineered *S.* Typhimurium during systemic infection in the myeloid compartment of the spleen. Given this, we decided to test the utility of the apoptotic backup pathways during oral infection. During infection with non-engineered WT *S.* Typhimurium, the bacteria naturally express both flagellin, SPI1 rod, and SPI1 needle in the gut lumen. Among these NLRC4 agonists, flagellin accounts for the vast majority of NLRC4 activation, and SPI1 rod and needle protein are relatively inefficiently detected (*Miao et al., 2006*). FliC<sup>ON</sup> engineering does not prevent the bacteria from expressing these endogenous genes. To create an infection model where the majority of NLRC4 activation arises only from the engineered FliC<sup>ON</sup>, we used a *flgB* mutant background to ablate endogenous flagellin expression and used a competitive index vs vector control bacteria. In the streptomycin-pretreated oral infection model, the IEC compartment can be assessed by examining gentamicin protected bacterial burdens in the cecum (*Rauch et al., 2017*), and from the same mice the myeloid compartment can be assessed in the draining mesenteric lymph nodes (MLN).

As a control, FliC<sup>ON</sup> burdens were equal to the vector control within the fecal samples, where no selective pressure should exist (*Figure 7A–B*). FliC<sup>ON</sup> was cleared from MLN after oral infection of WT mice (*Figure 7A–B*), which agrees with our above data from the spleen after intraperitoneal infection. FliC<sup>ON</sup> was also cleared in the gentamicin-protected cecal compartment in WT mice (*Figure 7A–B*), suggesting that the pyroptotic pathway leads to clearance from both IEC and macrophage compartments. Surprisingly, when we examined the apoptotic backup pathway by infecting *Gsdmd*<sup>-/-</sup> mice with FliC<sup>ON</sup>, we observed different results in the two compartments. There was no clearance of FliC<sup>ON</sup> in the MLN of *Gsdmd*<sup>-/-</sup> mice (*Figure 7A–B*), in agreement again with our data from the spleen. However, FliC<sup>ON</sup> was cleared in the gentamicin-protected cecal compartment of *Gsdmd*<sup>-/-</sup> mice with efficiencies that were equal to that seen in WT mice (*Figure 7A–B*). As a control, *Nlrc4*<sup>-/-</sup> mice were unable to clear FliC<sup>ON</sup> *S.* Typhimurium from any compartment (*Figure 7—figure supplement 1A–B*). Interestingly, *Casp1*<sup>-/-</sup> mice that retain only the ASC to caspase-8 backup pathway had reduced clearance of FliC<sup>ON</sup> *S.* Typhimurium from the cecum (*Figure 7—figure supplement 1A–B*). This suggests that the apoptotic backup pathways are specifically useful in IECs, in agreement with the data from Rauch et al., however, these same pathways fail in myeloid compartments in the lymph nodes or spleen.

We also investigated BID<sup>ON</sup> *S.* Typhimurium during oral infection again on a *flgB* background and found BID<sup>ON</sup> was also cleared from the cecum (*Figure 7C–D*). In agreement with our in vitro data showing that BID<sup>ON</sup>-induced apoptosis is slower and less efficient than the apoptotic backup pathways induced by FliC<sup>ON</sup> in *Gsdmd*<sup>-/-</sup> BMMs (*Figure 4A*), clearance of BID<sup>ON</sup> in the cecum was less efficient than clearance of FliC<sup>ON</sup> in *Gsdmd*<sup>-/-</sup> mice (*Figure 7A–D*). Interestingly, in the MLN compartment, BID<sup>ON</sup> appears to have a slight advantage versus the vector control *S.* Typhimurium (*Figure 7C–D*).

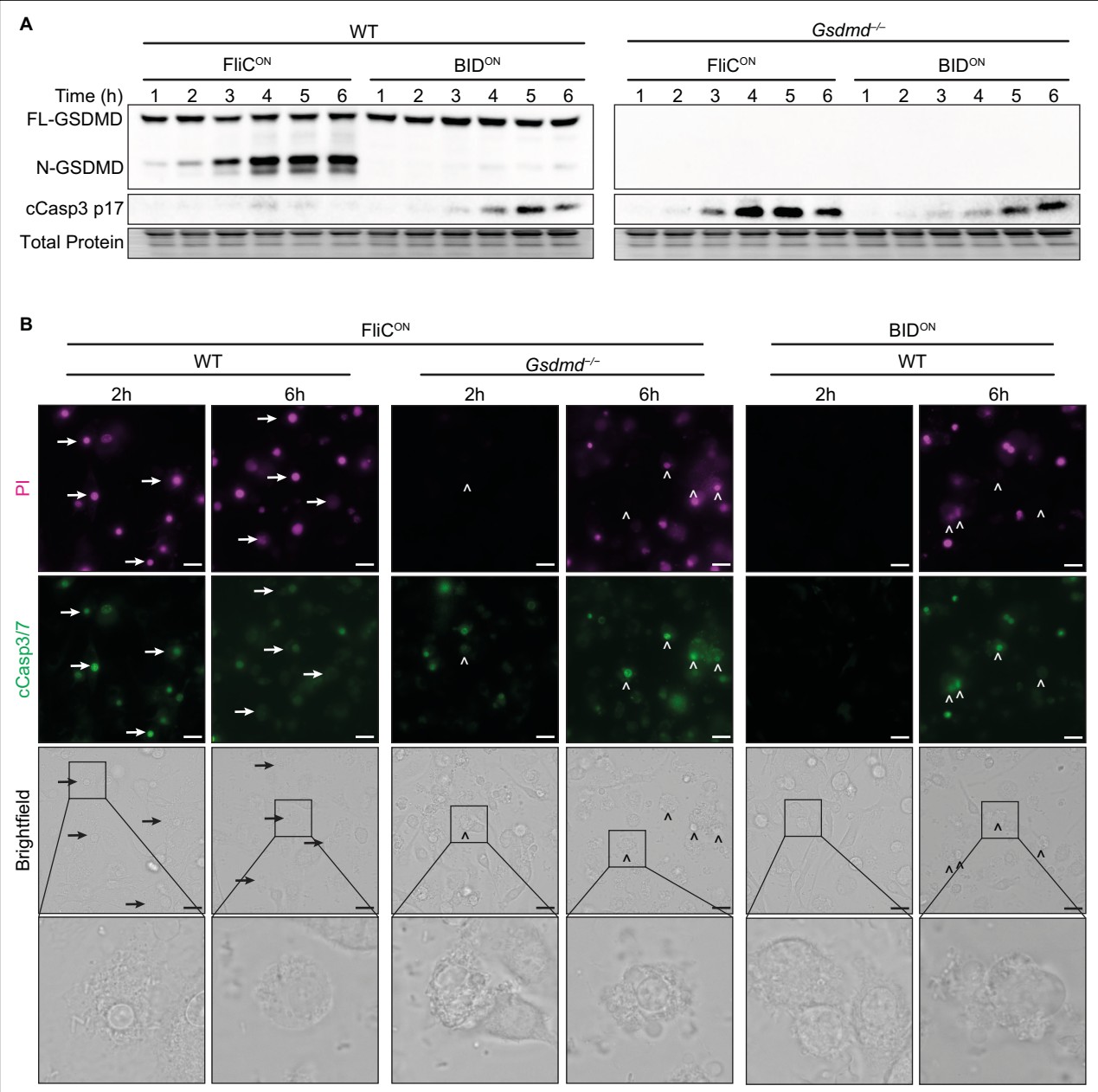

**Figure 4.** Apoptosis is induced slower than pyroptosis. (**A–B**) Bone marrow-derived macrophages (BMMs) were infected with indicated SPI2-induced *S.* Typhimurium strains. (**A**) Western blot analysis of whole cell lysates. Representative of five independent experiments. (**B**) Immunofluorescence and brightfield. Cells were stained with PI, cleaved caspase-3/7, and Hoechst and imaged at indicated timepoints. Representative image from three (brightfield, PI) or one (cleaved caspase-3/7) independent experiments. Z-stack of the 6 hr timepoint is represented in *Figure 1—video 1* and *Figure 3—video 1*. Z-stack slice 19 (FliC^ON in *Gsdmd*^−/−) and slice 20 (BID^ON in WT) shown here. 60 x magnification, scale bar 20 µm. Arrows, pyroptotic cells. Carrots, apoptotic cells.

The online version of this article includes the following source data and figure supplement(s) for figure 4:

**Source data 1.** Western blot images for *Figure 4A*.

**Figure supplement 1.** Vector control *S.* Typhimurium does not cause regulated cell death (RCD) in vitro.

This matches a weak, but repeatable trend seen in the splenic compartment in previous infections (*Figure 5A–B*, *Figure 7—figure supplement 1C–D*). Therefore, cell-intrinsic apoptotic pathways can also clear engineered *S.* Typhimurium from the IEC compartment, but not from lymph nodes or the spleen (*Figure 7E*).

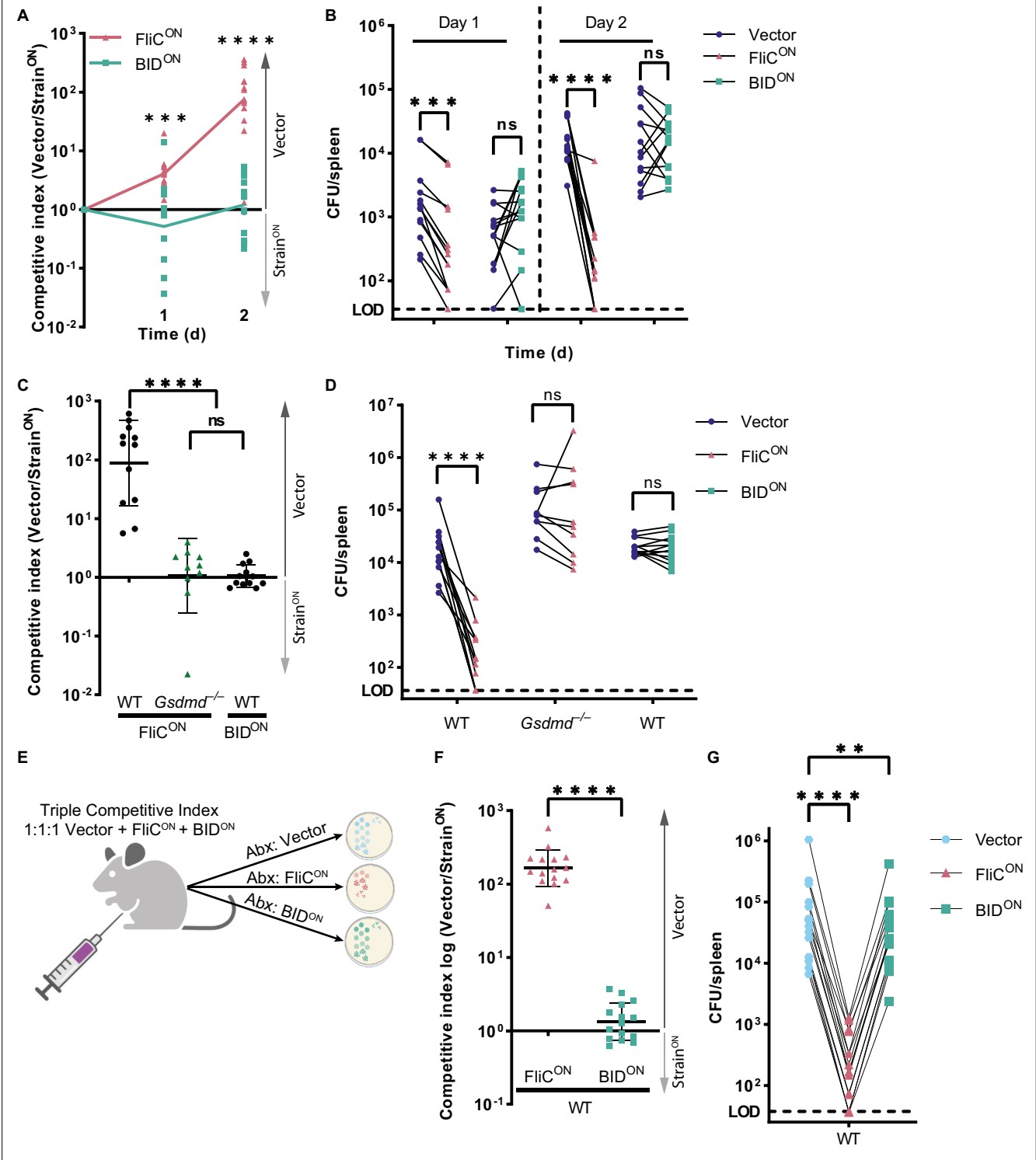

**Figure 5.** Intrinsic apoptosis does not clear engineered *S*. Typhimurium in the spleen. (**A–D**) Mice were infected with a 1:1 ratio of either FliC<sup>ON</sup> or BID<sup>ON</sup> and a vector control *S*. Typhimurium. Mice were infected with $5 \times 10^2$ CFU of each strain. Bacterial burdens in the spleen were determined at the indicated timepoints. (**A**) Timecourse competitive index infection in WT mice. Ratio of vector to either FliC<sup>ON</sup> or BID<sup>ON</sup> is graphed. Data is combined from three independent experiments, line connects means, n=13–14 mice per condition. Two-way ANOVA *** p<0.001, ****p<0.0001. (**B**) Individual burdens of vector and FliC<sup>ON</sup> or BID<sup>ON</sup> from (**A**). Paired vector and FliC<sup>ON</sup> or BID<sup>ON</sup> data from each mouse are connected by a line. Two-way repeated measure ANOVA n.s. p>0.05; ***p<0.001, ****p<0.0001. (**C**) Competitive index infection of indicated mice infected with either FliC<sup>ON</sup> or BID<sup>ON</sup>. Ratio of vector to either FliC<sup>ON</sup> or BID<sup>ON</sup> is graphed. Bacterial burdens in the spleen were determined at 48 hpi. Data is combined from three independent

*Figure 5 continued on next page*

*Figure 5 continued*

experiments, line representing mean ± SD, n=10–12 mice per condition. One-way ANOVA n.s. p>0.05, ****p<0.0001. (**D**) Individual burdens of vector and FliC[ON] or BID[ON] from (**C**). Paired vector and Strain[ON] data from each mouse are connected by a line. Two-way repeated measure ANOVA n.s. p>0.05, ****p<0.0001. (**E**) Schematic of triple competitive index model. (**F–G**) Mice were infected simultaneously with three strains, $5 \times 10^2$ CFU each of Cam[R] vector, Kan[R] FliC[ON], and Amp[R] BID[ON] *S.* Typhimurium. Bacterial burdens in the spleen were determined at 48 hpi. (**F**) Triple competitive index infection of WT mice. Ratio of Cam[R] vector to Kan[R] FliC[ON] or Amp[R] BID[ON] is graphed. Data is combined from three independent experiments, line representing mean ± SD, n=15. Unpaired two-tailed t-test ****p<0.0001. (**G**) Individual burdens of Cam[R] vector, Kan[R] FliC[ON], and Amp[R] BID[ON] from (**F**). Paired vector, FliC[ON], and BID[ON] data from each mouse are connected by a line. One-way repeated measure ANOVA **p<0.01, ****p<0.0001.

The online version of this article includes the following source data and figure supplement(s) for figure 5:

**Source data 1.** Data for *Figure 5A–D and F–G*.

**Figure supplement 1.** pWSK229 can be used for triple competitive index infection in vivo.

**Figure supplement 1—source data 1.** Data for *Figure 5—figure supplement 1A–B*.

## Discussion

There are many, interconnected pathways which cells utilize during regulated cell death. Such backup pathways may be useful in a situation where a pathogen has evolved to inhibit the primary mode of RCD. Here, we demonstrate that pyroptosis and apoptosis are not equivalent in their ability to clear engineered *S.* Typhimurium infection. We hypothesize that clearance after RCD is cell-type-dependent because of what happens both before and after cell death occurs. Once a cell initiates cell death signaling, it 'knows' that it will die (or rather, evolution has created signaling cascades that are predicated upon the initiation of RCD). However, each cell has distinct goals that it needs to complete before it loses all functional capacity, which we recently proposed to be termed 'bucket lists' (*Nozaki and Miao, 2023*). Successful completion of such bucket lists have beneficial effects after the cell dies. Bucket lists can be different in distinct cell types as well as in different signaling pathways in the same cell type. In the case of *S.* Typhimurium, the bacterium targets two distinct cell types: IECs and macrophages. After an IEC dies, it must ensure that it does not leave behind a hole in the epithelial

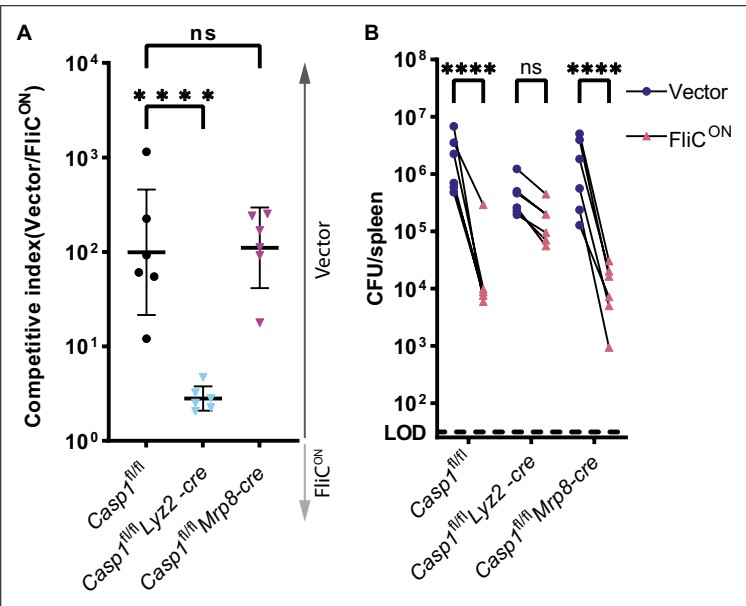

**Figure 6.** Pyroptosis clears FliC[ON] from myeloid compartment in vivo. (**A**) Mice were infected with $5 \times 10^4$ CFU of each strain. Bacterial burdens in the spleen were determined at 48 hpi. Ratio of vector to FliC[ON] is graphed. Combined two independent experiments, line representing mean ± SD, n=6 mice per genotype. One-way ANOVA n.s. p>0.05; ****p<0.0001. (**B**) Individual burdens from (**A**). Paired vector and FliC[ON] data from each mouse are connected by a line. Two-way repeated measure ANOVA. n.s. p>0.05, ****p<0.0001.

The online version of this article includes the following source data for figure 6:

**Source data 1.** Data for *Figure 6A–B*.

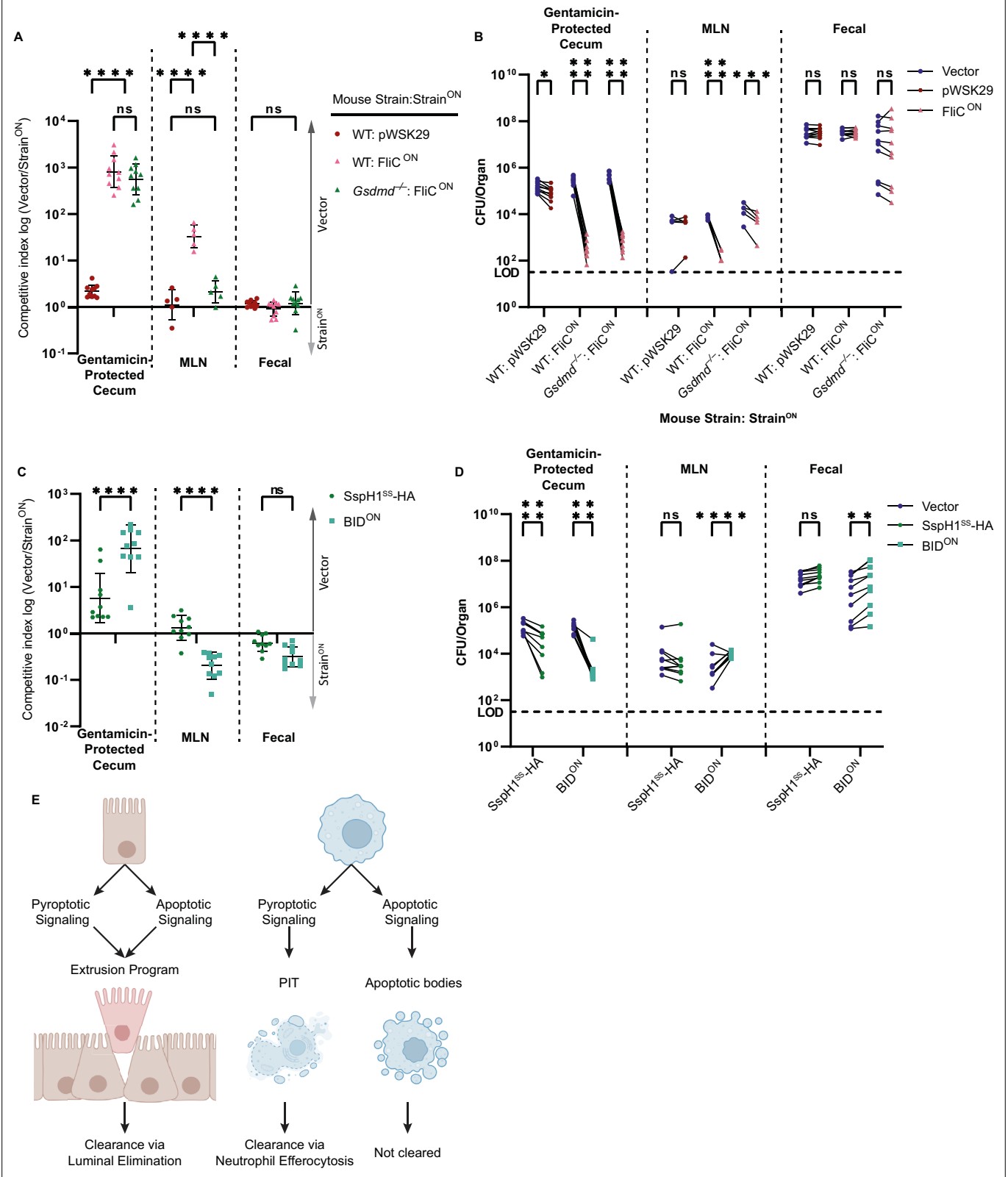

**Figure 7.** Apoptotic pathways lead to clearance in the cecum. (**A–D**) Mice were orally treated with 20 mg streptomycin, and 24 hr later orally infected with 1 × 10⁷ CFUs total bacteria comprised of a 1:1 ratio of the indicated ampicillin-resistant strain and kanamycin-resistant vector (pWSK129) control *S.* Typhimurium, all on a *flgB* mutant background. Bacterial burdens in the cecum, mesenteric lymph nodes (MLN), and fecal samples were determined at 48 hpi. (**A**) Competitive index is graphed as a ratio of vector to either pWSK29 or FliC^ON. Data is combined from two (cecum, fecal) or one (MLN)

*Figure 7 continued on next page*

*Figure 7 continued*

independent experiments (MLN was not harvested in the first experiment, where we harvested the spleen, which had negligible burdens; one additional representative experiment is shown in *Figure 7—figure supplement 1A–B*), line representing mean ± SD, n=10 (cecum, fecal) or 5 (MLN) mice per condition. Two-way repeated measure ANOVA n.s. p>0.05, ****p<0.0001. (**B**) Individual burdens of vector and pWSK29 or FliC[ON] from (**A**). Paired vector and Strain[ON] data from each mouse are connected by a line. Two-way repeated measure ANOVA n.s. p>0.05, *p<0.05, ***p<0.001, ****p<0.0001. (**C**) Competitive index infection of WT mice infected with either SspH1[SS]-HA or BID[ON]. Ratio of vector to either SspH1[SS]-HA or BID[ON] is graphed. Data is combined from two independent experiments, line representing mean ± SD, n=10 mice per condition. Two-way repeated measure ANOVA n.s. p>0.05, ****p<0.0001. (**D**) Individual burdens of vector and SspH1[SS]-HA or BID[ON] from (**C**). Paired vector and Strain[ON] data from each mouse are connected by a line. Two-way repeated measure ANOVA n.s. p>0.05, **p<0.01, ****p<0.0001. (**E**) Schematic demonstrating the ability of pyroptotic or apoptotic signaling to lead to clearance of engineered *S.* Typhimurium in either intestinal epithelial cells (IECs) or macrophages.

The online version of this article includes the following source data and figure supplement(s) for figure 7:

**Source data 1.** Data for *Figure 7A–D*.

**Figure supplement 1.** Clearance of FliC[ON] in the cecum is NLRC4-dependent.

**Figure supplement 1—source data 1.** Data for *Figure 7—figure supplement 1A–D*.

barrier. The bucket list of an IEC is relatively similar after pyroptotic or apoptotic signaling is initiated, as both pathways trigger the extrusion program (*Rauch et al., 2017*). Regardless of the initiating signaling events, an IECs key goal of maintaining barrier integrity after death remains the same. In contrast, a macrophage is not a barrier cell and, therefore, has no such bucket list task. Instead, in macrophages pyroptosis and apoptosis result in two different bucket lists. Pyroptosis converts a cell into pore-induced intracellular traps (PITs), whereas apoptosis converts a macrophage into apoptotic bodies, which are morphologically and immunologically distinct structures (*Elliott and Ravichandran, 2016*; *Jorgensen et al., 2016b*).

The bucket list of an IEC is centered around two goals: to remove the affected IEC from the body meanwhile maintaining epithelial barrier integrity. IECs accomplish both goals via a process called extrusion. After initiating either pyroptotic or apoptotic caspases, rather than immediately undergoing classic 'pyroptosis' or 'apoptosis,' an IEC first disassembles its cytoskeleton, then reassembles an actomyosin contractile ring to squeeze itself out of the monolayer. Neighboring cells simultaneously constrict beneath the dying IEC as it extrudes, maintaining barrier integrity (*Blander, 2018*). The extruded IEC is thereby shed from the body. Using non-engineered WT *S.* Typhimurium during oral infection, Sellin et al., show that NLRC4 reduces bacterial burdens in the cecum by triggering IEC extrusion (*Sellin et al., 2014*). Rauch et al. showed that these IECs can extrude via either the pyroptotic or backup apoptotic pathways (*Rauch et al., 2017*). Similarly, either pyroptotic or apoptotic signaling can reduce *Shigella flexneri* burdens from IECs during oral infection (*Roncaioli et al., 2023*). These conclusions are identical to the conclusion we draw from the FliC[ON] and BID[ON] engineered *S.* Typhimurium models. We propose that both pyroptotic signaling via caspase-1 and gasdermin D as well as backup apoptotic signaling via caspases-8/9/3 all converge upon similar bucket list tasks that drive similar IEC extrusion programs. It should be noted that pyroptotic signaling in IECs also induces the release of IL-18 and eicosanoids as part of the pyroptotic bucket list (*Müller et al., 2016*; *Rauch et al., 2017*). In regard to *S.* Typhimurium infection, these additional signaling molecules influence *S.* Typhimurium dissemination only at 72 hpi (*Müller et al., 2016*). Overall, extrusion driven by either pyroptotic or apoptotic pathways is sufficient to reduce both engineered and non-engineered *S.* Typhimurium burdens in the IEC compartment within 48 hpi.

More recently, Hausmann et al. showed that this IEC extrusion is not only beneficial to reduce the infected IEC compartment, but also restricts dissemination of *S.* Typhimurium to the draining MLN (*Hausmann et al., 2020*). This detection only occurred in IECs and not in macrophages in the MLN because non-engineered WT *S.* Typhimurium only expresses flagellin and SPI1 rod/needle during the invasion of IECs. After dissemination to macrophages, the non-engineered bacteria have successfully repressed flagellin expression and switched from SPI1 to SPI2 T3SS, thereby evading NLRC4 detection (*Hausmann et al., 2020*). In contrast, flagellin-engineered FliC[ON] prevents this evasion, triggering NLRC4-dependent pyroptosis. Using FliC[ON], we previously showed that after pyroptosis, the dead macrophage turns into a pore-induced intracellular trap (PIT) that traps *S.* Typhimurium in the cell corpse within which it is subsequently efferocytosed by neutrophils (*Jorgensen et al., 2016a*; *Jorgensen et al., 2016b*). Neutrophil ROS is then ultimately responsible for killing FliC[ON], as mice deficient in phagocyte oxidase are no longer able to clear FliC[ON] after pyroptosis (*Jorgensen et al.,*

*2016b*; *Miao et al., 2010a*). Therefore, there are at least two tasks on the bucket list of a pyroptotic macrophage: to trap bacteria within the PIT and to recruit neutrophils for efferocytosis. Pyroptosis also has other consequences that have not yet been fully proven to be essential bucket list tasks, including damaging intracellular bacteria (*Jorgensen et al., 2016b*; *Liu et al., 2016*; *Thurston et al., 2016*), releasing cytosolic molecules including nutrients (*Kovacs and Miao, 2017*), disrupting mitochondrial membranes to produce ROS (*de Torre-Minguela et al., 2021*; *Valenti et al., 2022*), and activating NINJ1 to cause membrane rupture (*Kayagaki et al., 2021*).

Apoptosis in macrophages, on the other hand, has a bucket list that is distinct from that of pyroptosis. An apoptotic macrophage packages itself into smaller apoptotic bodies which are then typically efferocytosed by other macrophages in what is classically an anti-inflammatory manner (*Morioka et al., 2019*; *Poon et al., 2014*). This bucket list requires careful disassembly and packaging of the internal cellular contents without losing membrane integrity, while also selectively releasing 'find me' and 'eat me' signals for the efferocytosing macrophages (*Elliott and Ravichandran, 2016*; *Morioka et al., 2019*; *Ravichandran, 2011*). In situations where our engineered *S.* Typhimurium induce apoptosis, they should be trapped within these apoptotic bodies (*Martin et al., 2012*). However, the apoptotic bucket lists tasks fail to clear our engineered *S.* Typhimurium from the macrophage compartment. This failure is not universal to all pathogens, as there is robust evidence that apoptosis successfully leads to the clearance of other intracellular bacteria (*Martin et al., 2012*). For example, non-engineered WT *Listeria monocytogenes* is cleared via apoptosis after cytotoxic T lymphocyte attack (*McDougal and Sauer, 2018*). Further, *L. monocytogenes* that have been engineered to directly induce intrinsic apoptosis (analogous to our BID^ON *S.* Typhimurium) is also cleared in the spleen and liver (*Theisen and Sauer, 2017*). Backup apoptosis is also successful in reducing bacterial replication in the lung during *Legionella pneumophila* infection (*Akhter et al., 2009*; *Gonçalves et al., 2019*). However, these apoptotic backup pathways are not successful in clearing other intracellular bacteria, similar to our results. The ASC to caspase-8 backup pathway was first reported in the context of *Francisella novicida* infection in caspase-1 deficient macrophages (*Pierini et al., 2012*). While the authors report that both pyroptosis and backup apoptosis limit bacterial replication in vitro, this did not translate to a difference in survival or bacterial burdens in a mouse infection (*Pierini et al., 2012*). Similarly, *C. violaceum* is cleared from the spleen via pyroptosis in WT mice, but backup apoptosis fails to clear *C. violaceum* in pyroptotic knockout mice (*Harvest et al., 2023*; *Maltez et al., 2015*; *Nozaki et al., 2022b*).

Why did apoptosis fail to clear our engineered *S.* Typhimurium in macrophages? We speculate that *S.* Typhimurium has virulence strategies that prevent apoptosis from being an effective clearance mechanism within the macrophage compartment. *S.* Typhimurium within apoptotic bodies might infect efferocytosing macrophages, preventing effective clearance analogous to the Trojan Horse from Greek mythology (*Giusiano, 2020*; *Liu et al., 2020*; *Poon et al., 2014*; *Saha et al., 2021*). Another possibility is the slower kinetics of apoptosis compared to pyroptosis permits bacterial replication before host cell death, and in this regard the longer retention of cytosolic nutrients within the apoptotic body may be beneficial to *S.* Typhimurium. Additionally, apoptotic cells can release specific metabolites that can be harvested by *S.* Typhimurium (*Anderson et al., 2021*). Adaptive immunity may change the consequences of apoptosis for *S.* Typhimurium in the macrophage compartment. Indeed, Doerflinger et al., used Δ*aroA S.* Typhimurium at 3 weeks post-infection and saw the beneficial utility of apoptotic signaling in clearing the bacteria (*Doerflinger et al., 2020*). Our engineered system is not amenable to studying later timepoints as overexpression of translocation signals can have detrimental effects that only manifest after 48 hpi (manuscript in press). Nevertheless, our findings show that apoptotic engineering fails to drive innate immune clearance against *S.* Typhimurium within macrophages. A final possibility is that our engineered strains are not successfully triggering apoptosis within splenic macrophages. This could be due to intrinsic differences between BMMs and splenic macrophages or could be due to bacterial virulence factors that fail to suppress apoptosis only in vitro. It is quite difficult to experimentally proves that apoptosis occurs in vivo due to rapid efferocytosis of the apoptotic cells. Any of these reasons could explain why apoptotic engineering did not result in clearance of the bacteria.

Here, we show that clearance of engineered *S.* Typhimurium after RCD is dependent upon the cellular compartment. Pyroptotic signaling leads to rapid clearance of engineered *S.* Typhimurium from both the intestinal and myeloid compartments, while apoptotic signaling leads to clearance only within the intestinal compartment (*Figure 7E*). Within IECs, pyroptotic and apoptotic signaling lead

to the same extrusion bucket list and, therefore, are equally sufficient in clearing both engineered and non-engineered *S.* Typhimurium (*Rauch et al., 2017*). This is likely due to the ability of extrusion to remove the infected cell and its intracellular bacteria from host tissues, forcing the bacteria to attempt invasion anew. *Salmonella* species are likely adapted to this invasion and re-invasion process, however, tipping the balance towards more efficient extrusion should benefit the host. Importantly, within the macrophage compartment, neither pyroptosis nor apoptosis intrinsically removes bacteria from the host tissue. Therefore, the unique bucket list during pyroptosis that recruits neutrophils to clear the bacteria is potentially superior to apoptosis (*Jorgensen et al., 2016b*). However, pyroptosis is largely evaded by non-engineered WT *S.* Typhimurium, which successfully replicates in macrophages (*Hausmann et al., 2020*; *Miao et al., 2010a*; *Miao et al., 2010b*; *Salcedo et al., 2001*). Therefore, we propose that it is the correct cellular bucket list in the context of each specific cell type, rather than simply the type of RCD, that ultimately leads to the clearance of intracellular bacteria.

# Materials and methods

## Key resources table

| Reagent type (species) or resource | Designation | Source or reference | Identifiers | Additional information |
|---|---|---|---|---|
| gene (*Mus musculus*) | *Bid* | NCBI | NM_007544.4 | AA79-102 (BH3 domain) used for plasmid construction |
| gene (*Salmonella enterica* serovar Typhimurium) | *SspH1* | GenBank | ACY87967.1 | AA1-137 (secretion signal) used for plasmid construction |
| strain, strain background (*Salmonella enterica* serovar Typhimurium, 14028s) | WT | Gift from Samuel I. Miller | | |
| strain, strain background (*Salmonella enterica* serovar Typhimurium, SL1344) | *flgB* | Gift from Kelly T. Hughes | | |
| strain, strain background (*Salmonella enterica* serovar Typhimurium, CS401) | *flgC ΔprgH-K* | Gift from Kelly T. Hughes | | |
| strain, strain background (*Mus musculus*, C57BL/6 J) | WT | Jax and Miao lab colony, Jax stock No. 000664 | | Colony bred WT mice were always used in experiments with colony bred knockout mice. Jax-purchased mice were only used in experiments having only WT mice from Jax. |
| strain, strain background (*Mus musculus*, C57BL/6 J) | *Mrp8-cre* | Miao lab colony, Jax stock No. 021614 | | |
| strain, strain background (*Mus musculus*, C57BL/6 J) | *Lyz2^{tm1(cre)Ifo}* (common name *LysM-cre*) | Miao lab colony, Jax stock No. 004781 | | |
| strain, strain background (*Mus musculus*, C57BL/6 J) | *Casp1^{fl/fl}* | Miao lab colony, *Hu et al., 2016* | | |
| strain, strain background (*Mus musculus*, C57BL/6 J) | *Casp1^{−/−}* | Miao lab colony, *Rauch et al., 2017* | | |
| strain, strain background (*Mus musculus*, C57BL/6 J) | *Casp1^{−/−} Casp11^{129mt/129mt}* (referred to as *Casp1/11^{−/−}*) | Miao lab colony, *Kuida et al., 1995* | | |
| strain, strain background (*Mus musculus*, C57BL/6 J) | *Gsdmd^{−/−}* | Miao lab colony, *Rauch et al., 2017* | | |
| strain, strain background (*Mus musculus*, C57BL/6 J) | *Nlrc4^{−/−}* | Miao lab colony, *Mariathasan et al., 2004* | | |

*Continued on next page*

*Continued*

| Reagent type (species) or resource | Designation | Source or reference | Identifiers | Additional information |
|---|---|---|---|---|
| strain, strain background (*Mus musculus*, C57BL/6 J) | *Bid*$^{-/-}$ | Miao lab colony, *Yin et al., 1999* | | |
| strain, strain background (*Mus musculus*, C57BL/6 J) | *Il1b/Il18*$^{-/-}$ | Miao lab colony, *Shornick et al., 1996*; *Takeda et al., 1998* | | |
| strain, strain background (*Mus musculus*, C57BL/6 J) | *Pycard*$^{-/-}$*Gsdmd*$^{-/-}$ (referred to as *Pycard/Gsdmd*$^{-/-}$) | Miao lab colony, crossed in this paper | | Produced by crossing *Pycard*$^{-/-}$ (also known as *Asc*$^{-/-}$) (*Mariathasan et al., 2004*) and *Gsdmd*$^{-/-}$ mice |
| strain, strain background (*Mus musculus*, C57BL/6 J) | *Bid*$^{-/-}$*Gsdmd*$^{-/-}$ (referred to as *Bid/Gsdmd*$^{-/-}$) | Miao lab colony, crossed in this paper | | Produced by crossing *Bid*$^{-/-}$ and *Gsdmd*$^{-/-}$ mice |
| antibody | Rabbit anti-cytochrome *c* monoclonal antibody | Cell Signaling Technology | 11940 | Western blot 1:750 dilution |
| antibody | Rabbit anti-GAPDH polyclonal antibody | Abcam | Ab9485 | Western blot 1:10,000 dilution |
| antibody | Rabbit anti-VDAC monoclonal antibody | Cell Signaling Technology | 4661 | Western blot 1:750 dilution |
| antibody | Rabbit anti-cleaved caspase-8 monoclonal antibody | Cell Signaling Technology | 8592 | Western blot 1:1000 dilution |
| antibody | Rat anti-BID monoclonal antibody | R&D | MAB860 | Western blot 1:500 dilution |
| antibody | Mouse anti-caspase-9 monoclonal antibody | Cell Signaling Technology | 9508 | Western blot 1:750 dilution |
| antibody | Rabbit anti-cleaved caspase-7 polyclonal antibody | Cell Signaling Technology | 9491 | Western blot 1:1000 dilution |
| antibody | Rabbit anti-cleaved caspase-3 polyclonal antibody | Cell Signaling Technology | 9661 | Western blot 1:750 dilution |
| antibody | Rabbit anti-gasdermin D monoclonal antibody | Abcam | Ab209845 | Western blot 1:1000 dilution |
| antibody | Mouse anti-HA.11 monoclonal antibody | Biolegend | MMS-101R | Western blot 1:2000 dilution |
| antibody | Goat anti-rabbit polyclonal antibody | Cell Signaling Technology | 7074 | Western blot secondary 1:2000 dilution |
| antibody | Goat anti-rat polyclonal antibody | Jackson ImmunoResearch | 112-035-062 | Western blot secondary 1:10,000 dilution |
| antibody | Goat anti-mouse polyclonal antibody | Jackson ImmunoResearch | 115-035-062 | Western blot secondary 1:10,000 dilution |
| recombinant DNA reagent | pWSK29 ("Vector") | *Wang and Kushner, 1991* | | See "Materials and methods, *Table 1*" |
| recombinant DNA reagent | pWSK129 ("Vector") | *Wang and Kushner, 1991* | | See "Materials and methods, *Table 1*" |
| recombinant DNA reagent | pDM001 ("FliC$^{ON}$") | *Miao et al., 2010a* | | See "Materials and methods, *Table 1*" |
| recombinant DNA reagent | pTA007 ("BID$^{ON}$") | This paper | | See "Materials and methods, *Table 1*" |
| recombinant DNA reagent | pTA021 ("SspH1$^{SS}$-HA") | This paper | | See "Materials and methods, *Table 1*" |

*Continued*

| Reagent type (species) or resource | Designation | Source or reference | Identifiers | Additional information |
|---|---|---|---|---|
| recombinant DNA reagent | pWSK229 ("Cam$^R$ Vector") | This paper | | See "Materials and methods, *Table 1*" |
| recombinant DNA reagent | pTA015 ("Kan$^R$ FliC$^{ON}$") | This paper | | See "Materials and methods, *Table 1*" |
| recombinant DNA reagent | pTA016 ("Kan$^R$ BID$^{ON}$") | This paper | | See "Materials and methods, *Table 1*" |
| commercial assay or kit | Pierce ECL | ThermoFisher Scientific | 32106 | |
| commercial assay or kit | SuperSignal West Pico PLUS ECL | ThermoFisher Scientific | 34580 | |
| commercial assay or kit | SuperSignal West Femto ECL | ThermoFisher Scientific | 34095 | |
| commercial assay or kit | CytoTox 96 LDH assay | Promega | G1780 | |
| software, algorithm | Prism 9 | GraphPad | | |
| Other | Hoechst 33342 | ThermoFisher | H3570 | Immuno-flourescence, used at 2 µg/ml |
| Other | Propidium Iodide | Sigma-Aldrich | P4864 | Immuno-flourescence, used at 1 µg/ml |
| Other | NucView-488 | Biotium | 10402 | Immuno-flourescence, used at 5 µM |

## Plasmid construction

pTA007 (BID$^{ON}$) was created by fusing the BH3 domain of murine BID (AA79-102) to the *S.* Typhimurium T3SS secretion signal of SspH1(AA1-137) via an HA tag. This fused SspH1$^{SS}$-HA-mBID$^{BH3}$ was put under the control of *sseJ* promotor and inserted into pWSK29. pTA021 (P$_{sseJ}$ *sspH1$^{SS}$-HA*) was created using the same strategy as pTA007, but did not include the murine BID$^{BH3}$ domain. pTA015 (Kan$^R$ FliC$^{ON}$) was created by digesting P$_{sseJ}$ *fliC fliS* from FliC$^{ON}$ and inserting the fragment into pWSK129. pTA016 (Kan$^R$ BID$^{ON}$) was created by digesting the P$_{sseJ}$ *sspH1$^{SS}$-HA-BID$^{BH3}$* from BID$^{ON}$ and inserting into pWSK129. pWSK229 was created by replacing the ampicillin resistance cassette of pWSK29 with a chloramphenicol resistance cassette cloned from pTwist Chlor MC from Twist Biosciences (South San Fransisco, CA).

## Bacterial strains and culture conditions

All *Salmonella enterica* serovar Typhimurium strains were derived from ATCC 14028s or SL1344. In vivo systemic infections were performed in 14028s with the indicated plasmids. In vivo oral infections were performed in SL1344 on a *flgB* mutant background with the indicated plasmids. In vitro infections were performed using *flgC ΔprgH-K S.* Typhimurium on the CS401 derivative of 14028s (used for the genetic deletion system) to eliminate flagellin and SPI1 T3SS expression from chromosomal loci. To eliminate flagellin expression from chromosomal loci during mouse infections, *flgB* 14028s *S.* Typhimurium were used for *Figure 2G–H , and* of three replicates of *Figure 5A–B*. However, we found

**Table 1.** Plasmids.

| Plasmids | Alias | Resistance | Notes | Reference |
|---|---|---|---|---|
| pWSK29 | Vector | Amp | Low copy vector | *Wang and Kushner, 1991* |
| pWSK129 | Vector | Kan | Low copy vector | *Wang and Kushner, 1991* |
| pDM1 | FliC$^{ON}$ | Amp | pWSK29 expressing *fliC fliS* from *sseJ* promoter | *Miao et al., 2010a* |
| pTA007 | BID$^{ON}$ or Amp$^R$ BID$^{ON}$ | Amp | pWSK29 expressing *sspH1$^{SS}$-HA-mBID$^{BH3}$* from *sseJ* promoter | This work |
| pTA021 | SspH1$^{SS}$-HA | Amp | pWSK29 expressing *sspH1$^{SS}$-HA* from *sseJ* promoter | This work |
| pWSK229 | Cam$^R$ Vector | Cam | Low copy vector | This work |
| pTA015 | Kan$^R$ FliC$^{ON}$ | Kan | pWSK129 expressing *fliC fliS* from *sseJ* promoter | This work |
| pTA016 | Kan$^R$ BID$^{ON}$ | Kan | pWSK129 expressing *sspH1$^{SS}$-HA-mBID$^{BH3}$* from *sseJ* promoter | This work |

that experimental results were identical using wild-type *S.* Typhimurium, therefore, wild type 14028s *S.* Typhimurium were used for all other systemic mouse infections. Plasmids are listed in *Table 1*. All strains were grown in 2 mL Miller's LB Broth (Apex Bioresearch Products, Houston, TX, Cat. 11–120) with appropriate antibiotics overnight at 37 °C with aeration. For the growth curve, a 1:1000 dilution of overnight culture into 50 ml LB broth was put at 37 °C with aeration. A 1 ml sample was removed and OD600 analyzed via spectrophotometer every hour until growth stabilized. An OD600 of $1 \approx 1 \times 10^9$ CFU was used.

## Macrophage culture and infection

BMMs were isolated as described (*Gonçalves and Mosser, 2015*) and confirmed mycoplasma negative via PCR. BMM media consisted of DMEM (ThermoFisher Scientific, Waltham, MA, Cat. 11995073) with 10% FBS and 15% LCM or 10 ng/ml M-CSF. Penicillin and streptomycin were added to the media during differentiation and thawing but withheld during seeding and infection. *flgC ΔprgH-K S.* Typhimurium was induced to express SPI2 by back diluting an overnight culture to OD600=0.026 in 3 mL SPI2-inducing media, then grown 16–20 hr in a 37 °C shaker (*Miao et al., 2002*). SPI2-induced bacteria were washed once with PBS prior to calculating MOI. For all BMM infections, 'Vector' control is pWSK29. Macrophages were infected with MOI 25, centrifuged for 5 min at $200 \times g$, incubated for 1 hr, then gentamicin (30 µg/ml) was added to the medium to kill any extracellular bacteria. Lipopolysaccharide (50 ng/ml) was added to normalize TLR activation for mock infected cells.

## Western blot analysis

For whole-cell lysates, macrophages were seeded at $2 \times 10^5$ cells/well in 24 well tissue culture treated plates the day before infection. Macrophages infected as described above. At indicated timepoints, media was aspirated, and 60 µl 1 x Laemmli Sample Buffer was added to lyse the cells. Samples were boiled for 5 min at 95 °C and frozen at –80 °C until analyzed. 12 µl sample was loaded into a 4–12% polyacrylamide TGX Stain-Free gel (Bio-Rad, Hercules, CA, Cat. 4568086) and run for 1 hr 15 min at 15 mA per gel. Gel was UV-activated in order to visualize the total protein. Protein was then transferred onto a 0.45 µm PVDF membrane (Millipore, Burlington, MA, Cat. IPFL85R), blocked with 5% non-fat dried milk in TBS plus 0.01% Tween (TBST) for 1 hr at room temperature, and incubated overnight at 4 °C with mild agitation in 5% milk in TBST plus indicated antibody: cytochrome *c* (1:750, rabbit, Cell Signaling Technology, Danvers, MA, Cat. 11940), GAPDH (1:10,000, rabbit, abcam, Cambridge, UK, Cat. Ab9485), VDAC (1:750, rabbit, Cell Signaling Technology, Cat. 4661), cleaved caspase-8 (1:1000, rabbit, Cell Signaling Technology, Cat. 8592), BID (1:500, rat, R&D Systems, Minneapolis, MN, Cat. MAB860), caspase-9 (1:750, mouse, Cell Signaling Technology, Cat. 9508), cleaved caspase-7 (1:1000, rabbit, Cell Signaling Technology, Cat. 9491), cleaved caspase-3 (1:750, rabbit, Cell Signaling Technology, Cat. 9661), gasdermin D (1:1000, rabbit, abcam, Cat. ab209845), or HA (1:2000, mouse, Biolegend, San Diego, CA, Cat. MMS-101R). Membranes were incubated for 1 hr at room temperature with appropriate secondary antibodies: goat anti-rabbit (1:2000, Cell Signaling Technology, Cat. 7074), goat anti-rat (1:10,000, Jackson ImmunoResearch, West Grove, PA, Cat. 112-035-062), or goat anti-mouse (1:10,000, Jackson ImmunoResearch, Cat. 115-035-062). ECLs used include Pierce ECL (ThermoFisher Scientific, Cat. 32106), SuperSignal West Pico PLUS (ThermoFisher Scientific, Cat. 34580), and SuperSignal West Femto (ThermoFisher Scientific, Cat. 34095). Images were taken using an Azure 500 Infrared Imaging System. For cell lysates, blots were probed in the following order: cleaved caspase-3, cleaved caspase-7, cleaved caspase-8, caspase-9, gasdermin D, BID, HA. For cytosolic and mitochondrial fractions, blots were probed in the following order: cytochrome *c*, VDAC, GAPDH, BID. Blots were stripped using a mild stripping buffer protocol (*Abcam, 2023*), then re-blocked with 5% milk. The p43 fragment of cleaved caspase-8 did not always fully strip in the apoptotic signaling conditions, leading to a faint p43 fragment seen in the gasdermin D blot even in *Gsdmd*$^{-/-}$ BMMs. Therefore, we do not believe this faint fragment represents an inactivating cleavage event of gasdermin D by caspase-3.

## Cytosolic and mitochondrial fraction isolation

Macrophages were seeded at $1 \times 10^6$ cells/well in 6-well non-TC treated plates (Genesee Scientific, Morrisville, NC, Cat. 25–100). Cells were infected as described above. At indicated timepoints, media was aspirated off and cells were lifted using 1 ml PBS +1 mM EDTA. Cells were spun down and washed

2x with sterile PBS at 300 × $g$ for 5 min. Cell pellets were permeabilized in 30 µl MOMP buffer (20 mM HEPES pH 7.4, 250 mM sucrose, 1 mM EDTA, 75 mM KCL, 2.5 mM MgCl$_2$, 0.05% digitonin) for 5 min on ice. Samples then spun at 15,000 × $g$ for 10 min at 4 °C. Supernatant was collected as a 'cytosolic' fraction. The pellet washed once in 30 µl PBS, and spun down again at 15,000 × $g$ for 10 min at 4 °C. Wash was discarded, and pellets lysed in 30 µl RIPA buffer on ice for 20 min. Samples spun final time at 18,000 × $g$ for 10 min at 4 °C, and supernatant was collected as a 'mitochondrial' fraction. 10 µl of 4x Laemmeli's Sample Buffer was added per sample, and then boiled at 95 °C for 5 min. Samples frozen at –80 °C until analyzed by SDS-PAGE and Western blot, described above. Protocol adapted from *Mahajan et al., 2014*.

## Cytotoxicity assay

Macrophages were seeded at $5 \times 10^4$ cells/well in a 96-well TC-treated plate. Cells were infected as described above. Supernatant was collected at indicated timepoints and frozen at –80 °C until analyzed. Cytotoxicity was determined by lactate dehydrogenase assay (CytoTox 96, Promega, Madison, WI, Cat. G1780). All samples and reagents were brought to room temperature before analyzing.

## Immunofluorescence and live cell microscopy

For microscopy experiments, bone marrow-derived macrophages were plated on 8 well chamber coverslips (ThermoFisher, Cat. 155409) at a density of $2.85 \times 10^5$ cells/cm$^2$. At the time of adding gentamicin 1 hr after initiating infection, the following immunofluorescent dyes were added. Dyes were used at the following concentrations: Hoechst 33342 at 2 µg/ml (ThermoFisher, Cat. H3570), Propidium Iodide at 1 µg/ml (Sigma-Aldrich, St. Louis, MO, Cat. P4864), and NucView-488 at 5 µM (Biotium, Fremont, CA, Cat. 10402). NucView-488 was selected to visualize activation of Caspase-3 in apoptosis, and Caspase-7 via Caspase-1 in pyroptosis. Images were captured using a Keyence BZ-X810 All-in-One Fluorescence Microscope using 20 x and 60 x objectives. For 20 x image stitching, 15 fields were captured and stitched using BZ-X 800 Image Analyzer 1.1.1.8 software.

## Mice and mouse infections

All mouse strains were bred and housed at Duke University in a pathogen-specific free facility. For infection mice were transferred to a BSL2 infection facility within Duke University, and mice allowed to acclimate for at least two days prior to infection. Wild type C57BL/6 J (Jackson Laboratory #000664), *Mrp8-cre* (Jackson Laboratory #021614), *Lyz2-cre* (Jackson Laboratory #004781), *Casp1$^{fl/fl}$* (*Hu et al., 2016*), *Casp1$^{-/-}$Casp11$^{129mt/129mt}$* (referred to as *Casp1/11$^{-/-}$*) (*Kuida et al., 1995*), *Gsdmd$^{-/-}$* (*Rauch et al., 2017*), *Nlrc4$^{-/-}$* (*Mariathasan et al., 2004*), *Casp1$^{-/-}$* (*Rauch et al., 2017*), *Bid$^{-/-}$* (*Yin et al., 1999*), *Il1b/Il18$^{-/-}$* (*Shornick et al., 1996*; *Takeda et al., 1998*). *Pycard$^{-/-}$Gsdmd$^{-/-}$* (referred to as *Pycard/Gsdmd$^{-/-}$*) were created by crossing *Pycard$^{-/-}$* (also known as *Asc$^{-/-}$*) (*Mariathasan et al., 2004*) and *Gsdmd$^{-/-}$* mice together. *Bid$^{-/-}$Gsdmd$^{-/-}$* (referred to as *Bid/Gsdmd$^{-/-}$*) were created by crossing *Bid$^{-/-}$* and *Gsdmd$^{-/-}$* mice together. Animal protocols were approved by the Institutional Animal Care and Use Committee (IACUC) at the University of North Carolina at Chapel Hill (under protocols 18–175.0 and 19–166.0) or by the IACUC at Duke University (under protocols A018-23-01 and A043-20-02) and met guidelines of the US National Institutes of Health for the humane care of animals. Details of sample size determination, randomization, and blinding can be found in the MDAR.

For competitive index infections, inoculum was composed equally of the vector control (pWSK129) and experimental strain (FliC$^{ON}$ or BID$^{ON}$, as indicated). For triple competitive index, the inoculum was composed equally of $5 \times 10^2$ CFU chloramphenicol-resistant vector control (pWSK229), Kan$^R$ FliC$^{ON}$, and Amp$^R$ BID$^{ON}$ for $1.5 \times 10^3$ total CFU. For systemic infection, mice were infected intraperitoneally with a total of $1 \times 10^3$ CFU (low dose) or $1 \times 10^5$ CFU (high dose) *S.* Typhimurium. Spleens were harvested at indicated timepoints and homogenized in a 2 ml homogenizer tube (Fisher Brand, Cat. 14-666-315) containing 1 ml sterile PBS and one 5 mm stainless steel bead (QIAGEN, Hilden, Germany, Cat. 69989). For oral infection, mice were first fasted for 4 hr and then orally inoculated with 20 mg of streptomycin sulfate (Sigma-Aldrich, Cat. 59137) which was filter sterilized using 0.22 µm filter (Genesee, Cat. 25–244). The next day, mice were again fasted for 4 hr and then orally infected with a total of $1 \times 10^7$ CFU *S.* Typhimurium. Food was returned 2 hr post-infection. At 48 hpi, the cecum, mesenteric lymph nodes, and a fecal sample were isolated. The fecal sample was homogenized under

the same condition as spleens, described above. The MLN was homogenized using a 2 mL homogenizer tube (OMNI International, Kennesaw, GA, Cat. 19–649) containing 1 ml sterile PBS and 7–10 2.4 mm stainless steel beads (OMNI International, Cat. 19-640-3). The cecum was first washed in PBS, then incubated on a room temperature rocker at 25 rpm in 400 ug/ml gentamycin (ThermoFisher, Cat. 15750060) for 30 min, then washed in fresh PBS for another 30 min. Caeca were then homogenized under the same conditions as MLNs. Spleens and fecal samples homogenized using a Retsch MM400 homogenizer for 5 min at 30 Hz. Fecal samples were spun down for 30 s and 5000 rpm after homogenizing. Cecum and MLNs were homogenized using a Fisherbrand Bead Mill 24 homogenizer at speed 4 for four cycles of 1 min homogenizing with 30 s dwell time. After homogenization, lysates were serially diluted 1:5 in sterile PBS and plated on LB plates containing appropriate antibiotics. Plates were incubated overnight at 37 °C and colony-forming units counted. If after harvest, zero CFUs were collected for all bacterial strains used in the inoculum, that a single mouse was considered 'uninfected,' which could occur due to experimental error, and was excluded from the study. If even one CFU was present for any of the bacterial strains used in the inoculum, that mouse was considered 'infected' and included in the study data. A total of four mice were excluded using these criteria: one *Nlrc4*$^{-/-}$ D1 mouse in *Figure 2B and D*, one D3 mouse from *Figure 2—figure supplement 1A–B*, and two mice from a single independent experiment in *Figure 5A–B* (one D2 BID$^{ON}$ mouse and one D1 FliC$^{ON}$ mouse). Competitive index results are presented as vector CFU/experimental CFU, normalized to the ratio of plated inoculum.

## Statistics

All statistical analysis was performed with GraphPad Prism 9. Discrete data was first assessed for normal distribution using a Shapiro-Wilk normality test. Data with normal distribution was analyzed with either an unpaired two-tail t-test (two groups) or a one-way ANOVA (three or more groups). Discrete data that did not have a normal distribution was analyzed with a Mann-Whitney (two groups) or Kruskal-Wallis (three or more groups). Experiments with two factors were analyzed with a two-way ANOVA. Detailed results for the statistical analyses are included as part of the Source Data file for each figure.

## Acknowledgements

We thank Richard Flavell, Russell Vance, and Vishva Dixit for sharing mice. This work is supported by NIH grants AI133236, AI139304, AI136920 (EAM), and AI133236-04S1(CKH), and the National Science Foundation Graduate Research Fellowship Program DGE 2139754 (TJA). Any opinions, findings, and conclusions or recommendations expressed in this material are those of the author(s) and do not necessarily reflect the views of the National Science Foundation. *Figure 7E* was created with BioRender.com under agreement number GZ25BXK3N9.

## Additional information

### Funding

| Funder | Grant reference number | Author |
| --- | --- | --- |
| National Institute of Allergy and Infectious Diseases | AI133236 | Edward A Miao |
| National Institute of Allergy and Infectious Diseases | AI139304 | Edward A Miao |
| National Institute of Allergy and Infectious Diseases | AI136920 | Edward A Miao |
| National Institute of Allergy and Infectious Diseases | AI133236-04S1 | Carissa K Harvest |
| National Science Foundation | Graduate Research Fellowship Program DGE 2139754 | Taylor J Abele |

| Funder | Grant reference number | Author |
|---|---|---|

The funders had no role in study design, data collection and interpretation, or the decision to submit the work for publication.

## Author contributions
Taylor J Abele, Conceptualization, Data curation, Formal analysis, Validation, Investigation, Methodology, Writing – original draft, Writing – review and editing; Zachary P Billman, Lupeng Li, Carissa K Harvest, Alexia K Bryan, Investigation; Gabrielle R Magalski, Joseph P Lopez, Conceptualization; Heather N Larson, Xiao-Ming Yin, Resources; Edward A Miao, Supervision, Funding acquisition, Writing – original draft, Project administration, Writing – review and editing

## Author ORCIDs
Taylor J Abele ⓘ http://orcid.org/0000-0002-7910-3381
Edward A Miao ⓘ https://orcid.org/0000-0001-7295-3490

## Ethics
Animal protocols were approved by the Institutional Animal Care and Use Committee (IACUC) at the University of North Carolina at Chapel Hill (under protocols 18-175.0 and 19-166.0) or by the IACUC at Duke University (under protocols A018-23-01 and A043-20-02) and met guidelines of the US National Institutes of Health for the humane care of animals.

Joint Public Review: https://doi.org/10.7554/eLife.89210.3.sa1
Author Response https://doi.org/10.7554/eLife.89210.3.sa2

# Additional files

## Supplementary files
• MDAR checklist

## Data availability
All study data are included in the article under source data files and have been uploaded to LabArchives in compliance with Duke University and NIH policies. Plasmid maps of novel constructs will be made available upon request and have been saved in Benchling in accordance with Duke University guidelines.

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
