## [Editor Report · eLife assessment]

Host cell death is an effective strategy to protect against infection, and is believed to function primarily by the elimination of the intracellular niche for pathogen replication. Abele and colleagues address an **important** question: does the mode of cell death affect its effectiveness in pathogen clearance? Consistent with prior observations, the authors provide **compelling** new evidence that the answer can depend on the cell type and/or tissue involved.

---

## [Referee Report · Joint Public Review]

In the present manuscript, Abele et al use *Salmonella* strains modified to robustly induce one of two different types of regulated cell death, pyroptosis or apoptosis in all cell types to assess the role of pyroptosis versus apoptosis in systemic versus intestinal epithelial pathogen clearance. They demonstrate that in systemic spread, which requires growth in macrophages, pyroptosis is required to eliminate *Salmonella*, while in intestinal epithelial cells (IEC), extrusion of the infected cell into the intestinal lumen induced by apoptosis or pyroptosis is sufficient for early pathogen restriction. The methods used in these studies are thorough and well controlled and lead to robust results, that mostly support the conclusions. The impact on the field is considered minor as the observations are somewhat redundant with previous observations and not generalizable due to cited evidence of different outcomes in other models of infection and a relatively artificial study system that does not permit the assessment of later timepoints in infection due to rapid clearance. This excludes the study of later effects of differences between pyroptosis and apoptosis in IEC such as i.e. IL-18 and eicosanoid release, which are only observed in the former and can have effects later in infection.

---

## [Author Response]

The following is the authors’ response to the original reviews.

**Reviewer #1 (Public Review):**
In the present manuscript, Abele et al use *Salmonella* strains modified to robustly induce one of two different types of regulated cell death, pyroptosis or apoptosis in all growth phases and cell types to assess the role of pyroptosis versus apoptosis in systemic versus intestinal epithelial pathogen clearance. They demonstrate that in systemic spread, which requires growth in macrophages, pyroptosis is required to eliminate *Salmonella*, while in intestinal epithelial cells (IEC), extrusion of the infected cell into the intestinal lumen induced by apoptosis or pyroptosis is sufficient for early pathogen restriction. The methods used in these studies are thorough and well-controlled and lead to robust results, that mostly support the conclusions. The impact on the field is considered minor as the observations are somewhat redundant with previous observations and not generalizable due to cited evidence of different outcomes in other models of infection and a relatively artificial study system that does not permit the assessment of later time points in infection due to rapid clearance. This excludes the study of later effects of differences between pyroptosis and apoptosis in IEC such as i.e. IL-18 and eicosanoid release, which are only observed in the former and can have effects later in infection.” We thank the reviewer for their time and effort in assessing our manuscript.

We agree with the reviewer’s overall assessment. One minor clarification is that the engineering used does not express the proteins in “all growth phases”, but rather only when the SPI2 T3SS is expressed; we used the sseJ promoter, which is a SPI2 effector.

**Reviewer #2 (Public Review):**
In this study, Abele et al. present evidence to suggest that two different forms of regulated cell death, pyroptosis and apoptosis, are not equivalent in their ability to clear infection with recombinant *Salmonella* strains engineered to express the pro-pyroptotic NLRC4 agonist, FliC ("FliC-ON"), or the pro-apoptotic protein, BID ("BID-ON"). In general, individual experiments are well-controlled, and most conclusions are justified. However, the cohesion between different types of experiments could be strengthened and the overall impact and significance of the study could be articulated better. ”

We thank the reviewer for their time and effort in assessing our manuscript. We agree with the reviewer’s overall assessment.

**Reviewer #1 (Recommendations For The Authors):**
Abstract: While new terms are sometimes useful for the visualization of concepts and I appreciate the "bucket list" analogy, it is not yet an accepted term in cell death research, and using it twice in the abstract seems out of order. ”

We opted to keep the term, but reduce its use to once in the abstract with a specific comment on the recent coining of the term: “We recently suggested that such diverse tasks can be considered as different cellular “bucket lists” to be accomplished before a cell dies.” We recently coined this term in a review in Trends in Cell Biology, where three reviewers had quite positive comments about the concept. Time will tell whether this is a useful term for the cell death field or not.

“In figure 2C-F Caspase 1 and Gsdmd deficient animals have higher levels of vector control strain than WT or Nlrc4. Could this be due to the redundancy with Nlrp3 in systemic infection described by Broz et al? Please mention in the description of the results.”

The reviewer correctly points out a trend in the data. However, our experiments are not powered to show that this difference is statistically significant. Nevertheless, we now make note of the trend, and cite prior papers that have observed NLRC4 and NLRP3 redundancy against non-engineered *S. Typhimurium* strains.

“The observation that apoptosis does not affect *Salmonella* systemically would be strengthened if the experiments using the BIDon strain could be taken out to a later time point, i.e. 72 or 96 h.”

Indeed, we wanted to extend our studies to these timepoints. However, although expression of the SspH1 translocation signal is benign for 48 h, by 72 h this causes mild attenuation (regardless of whether the BID-BH3 domain is attached as cargo). We think that the degree of difficulty for SPI2 effectors to reprogram the vacuole increases over time, and that only beyond 48 h does SPI2 need to function at peak efficiency. This observation will be reported in a second manuscript that is written and will be submitted within this month. We are happy to supply this manuscript to reviewers if they would like to see the results. We also added text to the discussion to alert the reader to the caveats of engineering *S. Typhimurium* at later timepoints.

“Discussion: The authors claim that pyroptotic and apoptotic signaling in IEC have the same outcome and IEC only has extrusion as a task. However, upon pyroptosis, IEC also releases IL-18 and eicosanoids, which is not the case during apoptosis. While the initial extrusion makes all the difference in early infection, Mueller et al 2016 showed that lack of IL-18 has an effect on *Salmonella* dissemination at a 72h time point. The FlicON model can not test later time points as the bacteria will be cleared by then, but this caveat should be discussed.”

We revised the text in the discussion to make it clear that extrusion is not the only bucket list item for IECs, and that IL-18 and eicosanoids are included in the bucket list for IECs after caspase-1 activation, and add the citation to Muller et al.

**Reviewer #2 (Recommendations For The Authors):**
1. The manuscript is written in a rather colloquial style. Additional editing is recommended. ”

We edited the abstract to limit the use of the bucket list term and to make more clear that this is a new term that our lab has proposed in a recent review in Trends in Cell Biology. The managing editor for the current manuscript at eLife commented that the prose was lively and thoughtful. We would be happy to make edits if the reviewer has more specific suggestions.

1. It is not obvious from the Results section that all mouse infections were, in fact, mixed infections. This should be stated more clearly. Additionally, there is a minor concern regarding in vivo plasmid loss over time.

We added text to the results to make this clearer at the beginning of each in vivo figure in the paper. Our experiments are intentionally blind to any *Salmonella* that have lost the plasmid. These bacteria essentially convert to a wild type phenotype, and thus are no longer representative of FliCON or BIDON bacteria. We also verify the long established equal competition between pWSK29 (amp) and pWSK129 (kan) in Figure 2—figure supplement 1A-B. Prior experiments from the laboratory of Sam Miller and others in the 1990s showed that plasmid loss occurs at a rate of less than 1%.

1. Results shown in Figure 4 are difficult to interpret. Essentially, the experiment is aimed at comparing the two engineered *Salmonella* strains (FliC-ON and BID-ON). However, these strains are very different from one another, which may have a confounding effect on the interpretation of the data.”

The reviewer has interpreted the experiment correctly. We wanted to make clear to the reader that the two strains induce apoptosis under different kinetics. Indeed, it would be very surprising if two different engineering methods created strains that caused apoptosis with identical kinetics. We make two text edits to the results to make this clearer, concluding with “Overall, both ways of achieving apoptosis are successful in vitro, but with slightly different kinetics.”.

1. What new insights into mechanisms of bacterial pathogenesis and host response are gained by using recombinant *Salmonella* (over)expressing a pro-apoptotic protein is not clearly stated.”

We modify the introduction to make this more clear, stating: “Here, we investigate whether apoptotic pathways could be useful in clearing intracellular infection. Because *S. Typhimurium* likely evades apoptotic pathways, we again use engineering in order to create strains that will induce apoptosis. This allows us to study apoptosis in a controlled manner in vivo.”

1. The Discussion section, while provocative, seems speculative and should be revised. Concepts of "backup apoptosis" and crosstalk between pyroptosis and apoptosis are intriguing, but it seems implausible to this reviewer that a cell might "know" that it will die, might "choose" how to die, and might aim to complete a "bucket list" before it loses all functional capacity. The usage of these types of terms does not help bolster the authors' central conclusions. ”

We agree that cells do not “choose” pathways for regulated cell death. We had over-anthropomorphized the concepts surrounding these interconnected cell death pathways that are created by evolution. We edited the introduction and discussion to remove the “choose” term. However, we kept the second phrase using “know” in the discussion with an added clarifier: “Once a cell initiates cell death signaling, it “knows” that it will die (or rather evolution has created signaling cascades that are predicated upon the initiation of RCD).”. Sometimes anthropomorphizing scientific concepts can be a useful tool to facilitate understanding of complex scientific concepts. For example, the “Red Queen hypothesis” clearly anthropomorphizes the concept of continuous evolution to maintain an evolutionary equilibrium. We have found that scientists in the cell death field often think that modes of cell death are or should be interchangeable. We hope that the idea of the “bucket list” will help to crystalize the idea that distinct processes leading up to different types of regulated cell death can have very different consequences during infection.

Additional Comments from the Reviewing Editor:1. The authors show that FliC-ON is not cleared from the spleen of Casp1 KO or Gsdmd KO mice. The conclusion is that the backup apoptosis pathways that should be present in these mice are insufficient to clear the bacteria from the spleen. However, although it is shown that bone marrow macrophages undergo apoptosis in vitro, I believe it is not shown that the apoptotic pathways are actually activated in the spleen. This seems like an important caveat. Could it be shown (or has it previously been shown) that the cells infected in the spleens of Casp1 KO or Gsdmd KO are activating apoptosis? If not, it seems possible that the reason the bacteria are not cleared is due to a lack of apoptosis activation rather than an ineffectiveness of apoptosis, and the authors could consider explicitly acknowledging this.”

We agree, and added to the discussion “A final possibility is that our engineered strains are not successfully triggering apoptosis within splenic macrophages. This could be due to intrinsic differences between BMMs and splenic macrophages or could be due to bacterial virulence factors that fail to suppress apoptosis only in vitro. It is quite difficult to experimentally prove that apoptosis occurs in vivo due to rapid efferocytosis of the apoptotic cells.”

1. Both reviewers were somewhat unhappy about some of the new terminology/metaphors that are introduced in the manuscript. I understand the reviewers' concerns but also feel that the writing is lively and thoughtful. It is up to the authors to decide whether to retain their new terminology, but the response of two expert reviewers might give the authors some pause. At a minimum, to address the concern about an unfamiliar term being used in the abstract, perhaps explicitly state that you are introducing "bucket list" as a new concept to help explain the results. The introduction of this concept may indeed be one of the novel contributions of the manuscript.”

We opted to keep the term, but reduce its use to once in the abstract with a specific comment on the recent coining of the term: “We recently suggested that such diverse tasks can be considered as different cellular “bucket lists” to be accomplished before a cell dies.” We recently coined this term in a review in Trends in Cell Biology, where three reviewers had quite positive comments about the concept. Time will tell whether this is a useful term for the cell death field or not.

1. Perhaps this is implied in the discussion already, but it might make sense to state the obvious difference between IECs and splenic macrophages which is that the death of the former results in the removal of the cell and its contents (i.e., *Salmonella*) from the tissue, whereas the death of the latter does not. This seems like the simplest explanation for why apoptosis restricts bacterial replication in IECs but not macrophages, and I am not sure if introducing the concept of a "bucket list" improves the explanation or not.”

We agree that this narrative nicely distills the differences between these cell types. We edited the final paragraph of the discussion to include this narrative.

1. Lastly, some minor comments-- p.2 "hyperactivate" instead of "hyperactive"?”

Corrected.

-- the authors may also want to mention Shigella, as it might provide another example that apoptotic C8dependent backup protects IECs”

Yes, indeed, this is a good comparison to make. We added this to the discussion.

-- p.8, in case readers are unfamiliar with the concept of a PIT, the authors should perhaps cite their own work when they first mention this concept (at the top of the page)”

Indeed, citation added.